# VLM2Vec-V2: Advancing Multimodal Embedding for Videos, Images, and Visual Documents

**Rui Meng[1][*][†]  Ziyan Jiang[2][*]  Ye Liu[1]  Mingyi Su[3]  Xinyi Yang[1]  Yuepeng Fu[4]**
**Can Qin[1]  Raghuveer Thirukovalluru[5]  Xuan Zhang[6]  Zeyuan Chen[1]  Ran Xu[1]**
**Caiming Xiong[1]  Yingbo Zhou[1]  Wenhu Chen[3]  Semih Yavuz[1]**

**[1]Salesforce Research    [2]UC Santa Barbara    [3]University of Waterloo**
**[4]Tsinghua University    [5]Duke University    [6]National University of Singapore**

**Reviewed on OpenReview:** `https://openreview.net/forum?id=TpU38jbKIJ`

`https://tiger-ai-lab.github.io/VLM2Vec/`

## Abstract

Multimodal embedding models have been crucial in enabling various downstream tasks such as semantic similarity, information retrieval, and clustering over different modalities. However, existing multimodal embeddings like VLM2Vec, E5-V, GME are predominantly focused on natural images, with limited support for other visual forms such as videos and visual documents. This limits their applicability in settings that require broader visual understanding, including AI agents, retrieval-augmented generation (RAG) systems, and recommendation. In this work, we propose VLM2Vec-V2, a unified framework for learning embeddings across diverse visual forms. To support comprehensive evaluation, we introduce MMEB-V2, an expanded benchmark that builds upon MMEB by adding five new task types: visual document retrieval, video retrieval, temporal grounding, video classification and video question answering. We then train VLM2Vec-V2, a unified embedding model capable of handling text, images, videos, and visual documents. The results demonstrate that VLM2Vec-V2 not only achieves strong performance on the newly introduced video and visual document tasks but also improves upon prior baselines on established image benchmarks. Our findings provide deeper insights into effective training strategies for unified embedding learning across diverse visual modalities.

## 1 Introduction

Embedding models play a crucial role in connecting data across various modalities. By encoding heterogeneous multimodal data into a shared dense representation space, they enable many down-stream applications like classification, clustering, retrieval. In recent years, we have witnessed significant advances in embedding models, largely driven by the progress of large foundation models. For instance, recent breakthroughs in text embedding (Su et al., 2023; Wang et al., 2024a; Meng et al., 2024; BehnamGhader et al.) have been achieved by integrating pretrained large language models with multi-task instruction embedding tuning. Similarly, Jiang et al.; Zhang et al. (2024); Chen et al. (2025) demonstrated strong performance across multiple text-image tasks by instruction-tuning multimodal large language models (MLLMs) into effective embedding models.

Existing multimodal embedding models are trained on datasets like MMEB (Jiang et al.) and M-BEIR (Wei et al., 2024), which are focused predominantly on natural images or photographs, sourced from MSCOCO (Lin et al., 2014), Flickr (Plummer et al., 2015) and ImageNet (Deng et al., 2009) datasets. These datasets fail to cover broader forms of visual information, such as documents, PDFs, websites, and videos.

---

[*]Contributed equally.
[†]Now at Google.

This limited coverage makes existing embedding models overly domain-specific and prevents them from generalizing well to realistic tasks such as document retrieval and YouTube video search. We empirically validate this claim in our experiments (Table 2). A model trained exclusively on visual documents, such as Colpali (Faysse et al.), achieves strong performance on document tasks (71.0%) but suffers a severe performance drop on image (34.9%) and video tasks (28.2%). Conversely, a model trained solely on image data, such as `VLM2Vec` (2B) (Jiang et al.), excels on image tasks (59.7%) but performs substantially worse on document (41.6%) and video tasks (29.0%).

To address these limitations, we introduce `MMEB-V2`, an advanced multimodal embedding dataset designed to train and evaluate embedding models across three key visual modalities: images, videos, and visual documents. Expanding on the original `MMEB` (Jiang et al.) framework, `MMEB-V2` broadens the evaluation scope to encompass five new tasks, including four video-based tasks—Video Retrieval, Moment Retrieval, Video Classification, and Video Question Answering — as well as one task centered on visual documents: Visual Document Retrieval. This comprehensive suite of tasks allows for robust assessment of multimodal embedding models across static, temporal, and structured visual data settings.

Built on top of `MMEB-V2`, we propose **`VLM2Vec-V2`**, a framework that adapts state-of-the-art multimodal large language models, such as Qwen2-VL (Wang et al., 2024b), into a unified embedding model. `VLM2Vec-V2` is trained using a mixture of instruction-following tasks spanning multiple task categories, enabling it to learn unified representations across a wide range of visual modalities.

In this paper, we adopt Qwen2-VL-2B as the base model for all experiments, balancing computational cost with the inclusion of video training data. Our framework is general and can be readily scaled to larger MLLMs.

The contributions of this work are threefold.

- We propose `MMEB-V2`, a comprehensive dataset for systematically evaluating embedding models on diverse tasks involving videos, images, and visual documents.

- We develop `VLM2Vec-V2`, a unified multimodal embedding framework that supports instruction following to produce general-purpose embeddings for a wide range of downstream tasks.

- Our experiments demonstrate that the 2B-parameter version of `VLM2Vec-V2` outperforms prior baselines across 78 datasets. Through detailed ablation studies, we identify effective training strategies for learning multimodal embedding models.

## 2 `MMEB-V2`: Expanding Embedding Benchmarks Beyond Image-Text

We introduce `MMEB-V2`, a comprehensive dataset designed to evaluate model performance on multimodal embedding tasks involving various combinations of text, image, video, and visual document modalities. In addition to the five task categories that involve natural images and text, `MMEB-V2` includes four video understanding tasks and one visual document understanding task. Figure 1 presents an overview of `MMEB-V2`.

Each task presents the model with a query and a corresponding set of candidate responses, where the goal is to select the correct target. The query consists of a combination of an instruction, a textual component, and a video or document. For video-based tasks, videos are represented as sequences of frames sampled at uniform intervals from the raw footage to ensure consistent temporal coverage. Instructions serve to specify the model's objective (e.g., "Recognize the category of the video contents."), and query texts can be questions, descriptions, or commands specific to the video (e.g., "How many red socks are above the fireplace at the end of this video?", "Select the clips of videos that contain a dolphin."). Each task is associated with a specific target type, which varies depending on the nature of the task. For example, in video classification, the model must identify the activity or object class label (e.g., "Yoga" or "Car").

To make the dataset practical and accessible for future research, we selectively downsample certain datasets to ensure that the full benchmark can be run within a reasonable amount of time. Our datasets span diverse domains, including sports, object recognition, daily-life activities, and movie or TV show clips. These

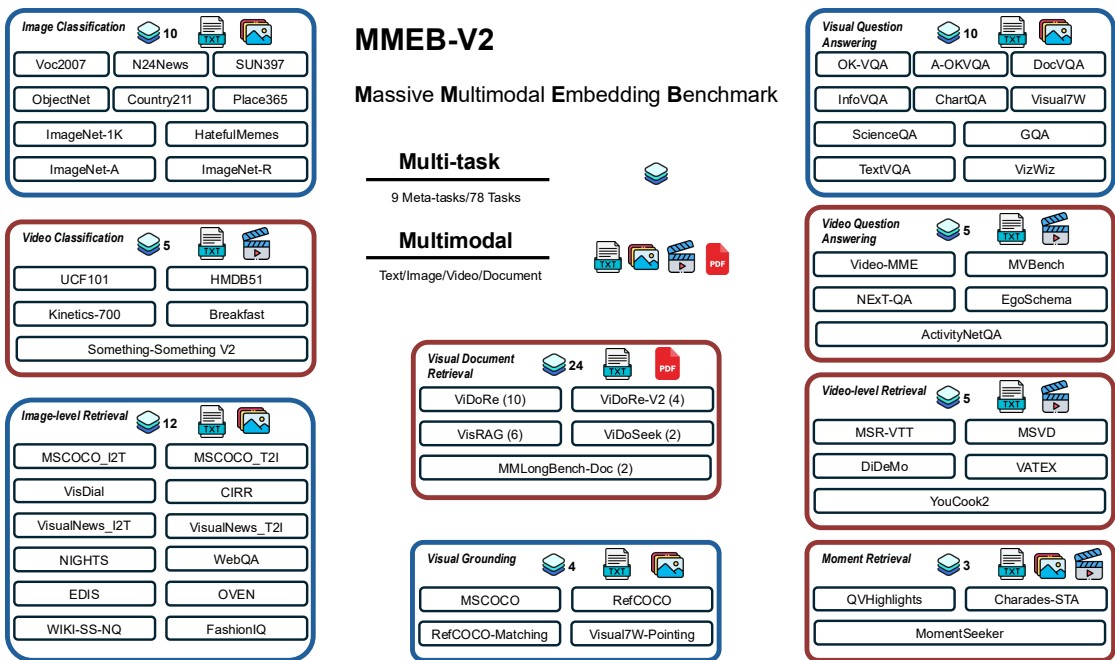

Figure 1: An overview of MMEB-V2, which includes 9 meta-tasks and 78 tasks in total. In addition to the original MMEB benchmark, MMEB-V2 introduces five new meta-tasks focused on video and visual documents. Tasks from MMEB are indicated with blue borders, while newly introduced tasks in MMEB-V2 are marked with red borders.

samples are drawn from varied sources such as YouTube, professional productions, and crowd-sourced content, ensuring both diversity and real-world relevance. Summary statistics for each task are presented in Table 1. Details about constructing each dataset are provided in Appendix A.1.

- **Video Retrieval (V-RET)** The query consists of an instruction, a descriptive text related to the video content, and a sequence of video frames. The model must retrieve the correct corresponding video from a pool of thousands of video candidates.
- **Moment Retrieval (M-RET)** The query consists of an instruction, a textual description, and optionally a full video, with the goal of retrieving the temporal segments that best matches the description. The model must select the ground-truth clip from approximately 10 candidate segments within the full video.
- **Video Classification (V-CLS)** Given an instruction and a sequence of video frames, the model is tasked with predicting the correct class label—related to the scene or action depicted—from a set of possible classes.
- **Video QA (V-QA)** The input consists of an instruction, a textual question, and a video. The model must select the correct answer from several options, including one correct choice and many distractors.
- **Visual Document Retrieval (VisDoc)** This task category evaluates the model's ability to retrieve structured visual documents—such as multi-page PDFs and slide decks—in response to natural language queries. We include five datasets in this benchmark. ViDoRe V1 & V2 (Faysse et al.; Macé et al., 2025) and VisRAG (Yu et al., 2024) are composed of multiple document QA datasets and cover a broad range of document types and use cases (e.g., charts and slides), though they were not originally designed for retrieval. To complement them, we include ViDoSeek (Wang et al., 2025) and MMLongBench-Doc (Ma et al., 2024), which provide fine-grained, page-level annotations suitable for retrieval evaluation. Besides, we reformat the two datasets to support both document-level and page-level evaluation.

## 3 Unified Embedding Model for Video, Image, and Visual Document

Built upon the VLM2Vec framework, VLM2Vec-V2 extends embedding learning to a broader range of modalities and tasks, which introduces additional challenges due to their structural and semantic characteristics. This

| Task | Query MOD | Target MOD | Domain | #Query | #Candidates |
|------|-----------|------------|--------|--------|-------------|
| **Video Retrieval (5 Tasks)** | | | | | |
| DiDeMo | T | V | Open | 1,004 | 1,004 |
| MSR-VTT | T | V | Open | 1,000 | 1,000 |
| MSVD | T | V | Open | 670 | 670 |
| VATEX | T | V | Open | 4,478 | 4,478 |
| YouCook2 | T | V | Cooking | 3,179 | 3,179 |
| **Moment Retrieval (3 Tasks)** | | | | | |
| QVHighlights | T + V | V | Vlog/News | 1,083 | 10 |
| Charades-STA | T + V | V | Activity | 727 | 10 |
| MomentSeeker | I + V | V | Open | 1,800 (1,602 deduped) | 10 |
| **Video Classification (5 Tasks)** | | | | | |
| Kinetics-700 | V | T | Open | 1,000 | 700 |
| SSv2 | V | T | Human-Object Interaction | 1,000 | 174 |
| HMDB51 | V | T | Open | 1,000 | 51 |
| UCF101 | V | T | Open | 1,000 | 101 |
| Breakfast | V | T | Cooking | 433 | 10 |
| **Video QA (5 Tasks)** | | | | | |
| MVBench | V + T | T | Spatial/Temporal | 4,000 | $3 \sim 5$ |
| Video-MME | V + T | T | Real-world | 2,700 | 4 |
| NExT-QA | V + T | T | Daily activity | 8,564 | 5 |
| EgoSchema | V + T | T | Egocentric | 500 | 5 |
| ActivityNetQA | V + T | T | Activity | 1000 | 2 |
| **Visual Document Retrieval (24 Tasks)** | | | | | |
| ViDoRe (10) | T | D | Documents | 280 - 1,646 | 70 - 999 |
| ViDoRe-V2 (4) | T | D | Documents | 52 - 640 | 452 - 1,538 |
| VisRAG (6) | T | D | Documents | 63 - 816 | 500 - 9,590 |
| ViDoSeek (2) | T | D | Documents | 1,142 | 5,349 |
| MMLongBench-Doc (2) | T | D | Documents | 838 | 6,492 |

Table 1: The statistics of `MMEB-V2`, which includes **42** tasks across five meta-task categories in addition to the original `MMEB`, are summarized below. Here, we list only the additional datasets introduced beyond those in `MMEB`. We consider four modalities (MOD): T (Text), I (Image), V (Video), and D (Visual Document).

section presents our methodology, beginning with a brief overview of the VLM2Vec framework (Section 3.1) and the foundation model selection (Section 3.2). Sections 3.3–3.4 then describe the key extensions beyond `VLM2Vec`, including our multimodal data mixing strategy (Section 3.3), and batching strategies to balance data sources during training (Section 3.4).

## 3.1 Preliminary: `VLM2Vec` Framework

As one of the earliest works to adapt MLLMs for universal multimodal embedding learning, `VLM2Vec` (Jiang et al.) formulates multimodal embedding learning as instruction-conditioned embedding alignment across diverse tasks and modalities. In `VLM2Vec`, each training example consists of a query–target pair $(q, t^+)$, where both $q$ and $t^+$ may comprise text, images, videos (extended in `VLM2Vec-V2`), or combinations thereof, together with a predefined task-specific instruction. Both $q$ and $t^+$ are independently passed through the same MLLM, and the final embeddings $\mathbf{h}_q$ and $\mathbf{h}_{t^+}$ are obtained from the last token's hidden state. Training is performed using a standard InfoNCE objective with in-batch negatives and optional hard negatives to align query and target representations, defined as:

$$\mathcal{L} = -\log \frac{\phi(\mathbf{h}_q, \mathbf{h}_{t^+})}{\phi(\mathbf{h}_q, \mathbf{h}_{t^+}) + \sum_{t^- \in \mathcal{N}(q)} \phi(\mathbf{h}_q, \mathbf{h}_{t^-})}, \quad \phi(\mathbf{h}_q, \mathbf{h}_t) = \exp\left(\frac{1}{\tau} \cos(\mathbf{h}_q, \mathbf{h}_t)\right) \tag{1}$$

where $\cos(\cdot, \cdot)$ denotes the cosine similarity function and $\tau$ is the temperature hyperparameter. Here, $t^-$ denotes a negative target sampled from the in-batch candidates or mined hard negatives associated with the query $q$. GradCache (Gao et al., 2021), a gradient caching technique, is used to increase the effective batch size for contrastive learning by decoupling backpropagation between the contrastive objective and the encoder, thereby removing encoder backward-pass dependencies along the batch dimension.

### 3.2 Foundation Model Selection

While VLM2Vec established an effective paradigm for transforming multimodal large language models into image–text embeddings, its design was primarily tailored to static visual inputs. In contrast, VLM2Vec-V2 extends this paradigm to support interleaved sequences of text, images, and videos, as well as long-form inputs such as multi-page visual documents and full-length videos.

Our goal is to learn a unified embedding space that generalizes across diverse visual modalities and tasks. This necessitates a backbone model that can flexibly encode interleaved multimodal sequences while scaling to long-context inputs. Based on these requirements, we adopt Qwen2-VL as the backbone of VLM2Vec-V2.

Qwen2-VL is particularly well suited to the challenges posed by VLM2Vec-V2, offering (i) Naive Dynamic Resolution for efficient processing of inputs with varying spatial resolutions, (ii) Multimodal Rotary Position Embedding (M-RoPE) to capture spatial and temporal structure in videos and visual documents, and (iii) a unified architecture that integrates 2D and 3D convolutions for consistent image and video understanding. Together, these capabilities enable VLM2Vec-V2 to effectively handle diverse and complex multimodal data.

### 3.3 Training Data and Multimodal Data Mixing

To train VLM2Vec-V2 effectively across diverse multimodal tasks, we curate a training dataset comprising three main sources: video-language instruction data, visual document retrieval, and image-based vision tasks.

First, we utilize training data from LLaVA-Hound (Zhang et al., 2025), which includes synthetic video-caption pairs and video QA examples generated by ChatGPT. Specifically, we use 300k video-caption pairs and 240k video QA pairs. For the caption data, we adopt two formats: using the caption as the query and video as the target in a video retrieval setup, or using the video as the query to retrieve the most relevant textual description from candidate captions. We denote these three video training datasets as Vid2Cap, Cap2Vid, and VideoQA in the following discussion.

Second, for visual document retrieval tasks, we incorporate datasets from ViDoRe (Faysse et al.) and VisRAG (Yu et al., 2024), including colpali train set (118k), VisRAG synthetic (239k), and VisRAG in-domain (123k), which provide training examples for image-based document understanding and retrieval. We denote these three video training datasets as ColPali, $VisRAG_{ind}$, and $VisRAG_{syn}$ in the following discussion.

Finally, we include image-text datasets from MMEB-train (Jiang et al.) to support generalization across a wide range of visual understanding tasks, including question answering, classification, retrieval, and visual grounding. These datasets help improve the robustness of the learned embeddings across multiple tasks.

We perform on-the-fly batch mixing guided by a pre-defined sampling weight table. This table specifies the relative probabilities of sampling from each dataset, enabling controlled exposure to different task types. By dynamically drawing samples during training, we ensure balanced coverage and prevent overfitting to any single modality or domain. We find that jointly mixing datasets from diverse sources and modalities leads to the most effective model training. Detailed ablation studies are presented in Section 4.5.1.

### 3.4 Batching Strategies

To support effective multi-task training over heterogeneous data sources, we introduce an *homogeneous sub-batching* strategy to enhance the hardness and stability of contrastive learning. Specifically, each full batch (e.g., size 1024) is divided into smaller sub-batches (e.g., 8 sub-batches of size 128), where each sub-batch is sampled independently from a single data source only. Compared to per-sample independent sampling, grouping similar samples into sub-batches increases intra-sub-batch homogeneity, which raises the difficulty of contrastive discrimination. At the same time, interleaving multiple such sub-batches preserves cross-task diversity within the full batch, avoiding the instability commonly observed in completely homogeneous batches that originate from a single source. This strategy strikes a balance between sample diversity and structural consistency, fostering more stable and robust optimization dynamics.

# 4 Experiments

## 4.1 Experiment Setting

### 4.1.1 Training Setting

We train `VLM2Vec-V2` using Qwen2-VL-2B-Instruct[1] (Wang et al., 2024b) as backbone, a batch size of 1,024 for 2K steps and a homogeneous sub-batch size of 64, with the loss temperature set to 0.02. To support scalable training, we use GradCache (Gao et al., 2021) to enable large global batch sizes and run all experiments on 8 H100 GPUs. For parameter-efficient training, we apply LoRA tuning with a rank of 16 and scaling factor $\alpha = 64$ using the PEFT framework (Mangrulkar et al., 2022). We use 8 uniformly sampled frames to represent each video during both training and evaluation.

Training `VLM2Vec-V2` with a 2B backbone on 8 NVIDIA H100 GPUs takes approximately 15 hours per 1k steps. Considering the training cost, all analyses in this work are conducted using the 2B backbone. Nevertheless, our framework is general and can be readily adapted to larger MLLMs, which may yield stronger embedding models.

### 4.1.2 Baselines

We compare against several VLM embedding models, including GME (Zhang et al., 2024), VLM2Vec (Jiang et al.), and LamRA (Liu et al., 2025), most of which are primarily trained on image-text pairs. While these models are not explicitly designed for video tasks, many can be adapted to handle video inputs by encoding multiple frames as sequential images. For video evaluation, GME and LamRA use a single middle frame, while the remaining models use 8 uniformly sampled frames.

In addition, to provide a fair comparison across modalities, we evaluate VLM2Vec-V2 against representative models specialized for each modality. Specifically, we include ColPali (Faysse et al.) (v1.3), a model tailored for document retrieval using a late interaction matching mechanism.

## 4.2 Evaluation Metrics

We use Hit@1 as the primary evaluation metric for all video and image tasks, measuring the proportion of queries where the correct target is ranked at the top. For visual document tasks, we report NDCG@5 to remain consistent with prior work in this domain.

## 4.3 Main Results

| Model | Image | | | | | Video | | | | | VisDoc | | | | | All |
| --- | --- | --- | --- | --- | --- | --- | --- | --- | --- | --- | --- | --- | --- | --- | --- | --- |
| | CLS | QA | RET | GD | Overall | CLS | QA | RET | MRET | Overall | VDRv1 | VDRv2 | VR | OOD | Overall | |
| # of Datasets → | 10 | 10 | 12 | 4 | 36 | 5 | 5 | 5 | 3 | 18 | 10 | 4 | 6 | 4 | 24 | 78 |
| **Baseline Models** | | | | | | | | | | | | | | | | |
| ColPali v1.3 (3B) | 40.3 | 11.5 | 48.1 | 40.3 | 34.9 | 26.7 | 37.8 | 21.6 | 25.5 | 28.2 | 83.6 | 52.0 | 81.1 | 43.1 | 71.0 | 46.0 |
| GME (2B) | 54.4 | 29.9 | 66.9 | 55.5 | 51.9 | 34.9 | 42.0 | 25.6 | 32.4 | 33.9 | 86.1 | 54.0 | 82.5 | 43.1 | 72.7 | 54.1 |
| GME (7B) | 57.7 | 34.7 | 71.2 | 59.3 | 56.0 | 37.4 | 50.4 | 28.4 | 38.2 | **38.6** | 89.4 | 55.6 | 85.0 | 44.4 | **75.2** | 57.8 |
| LamRA-Qwen2 (7B) | 59.2 | 26.5 | 70.0 | 62.7 | 54.1 | 39.3 | 42.6 | 24.3 | 34.6 | 35.2 | 22.0 | 11.5 | 37.4 | 21.0 | 23.9 | 40.4 |
| LamRA-Qwen2.5 (7B) | 51.7 | 34.1 | 66.9 | 56.7 | 52.4 | 32.9 | 42.6 | 23.2 | 37.6 | 33.7 | 56.3 | 33.3 | 58.2 | 40.1 | 50.2 | 47.4 |
| VLM2Vec-Qwen2VL (2B) | 58.7 | 49.3 | 65.0 | 72.9 | 59.7 | 33.4 | 30.5 | 20.6 | 33.0 | 29.0 | 49.8 | 13.5 | 51.8 | 33.5 | 41.6 | 47.0 |
| VLM2Vec-Qwen2VL (7B) | 62.7 | 56.9 | 69.4 | 82.2 | **65.5** | 39.1 | 30.0 | 29.0 | 40.6 | 34.0 | 56.9 | 9.4 | 59.1 | 38.1 | 46.4 | 52.3 |
| **Ours** | | | | | | | | | | | | | | | | |
| `VLM2Vec-V2` (2B) | 62.9 | 56.3 | 69.5 | 77.3 | 64.9 | 39.3 | 34.3 | 28.8 | 38.5 | 34.6 | 75.5 | 44.9 | 79.4 | 39.4 | 65.4 | **58.0** |

Table 2: Performance comparison between baseline models and our `VLM2Vec-V2` across image, video, and visual document tasks. CLS: classification, QA: question answering, RET: retrieval, GD: grounding, MRET: moment retrieval, VDR: ViDoRe, VR: VisRAG, OOD: out-of-domain.

---

[1] https://huggingface.co/Qwen/Qwen2-VL-2B-Instruct

Table 2 presents a comprehensive comparison between `VLM2Vec-V2` and a diverse set of baseline models across 78 datasets covering image, video, and visual document tasks. Full results are detailed in Appendix A.3. `VLM2Vec-V2` achieves the highest overall average score (58.0), outperforming multiple strong baselines, including GME, LamRA and `VLM2Vec`, which were built on the same Qwen2-VL backbone. This highlights the effectiveness of our unified training approach in delivering strong and balanced performance across different modalities and tasks. On image tasks, `VLM2Vec-V2` shows strong results, outperforming most baselines by a large margin and achieving performance comparable to `VLM2Vec`-7B despite being only 2B in size. For video tasks, it achieves competitive performance despite being trained on a relatively small amount of video data. In visual document tasks, `VLM2Vec-V2` outperforms all `VLM2Vec` variants, while still trailing behind ColPali, which is specifically optimized for VisDoc tasks.

### 4.4 Qualitative Analysis

To more comprehensively demonstrate improvements over baseline models, we perform qualitative comparisons between `VLM2Vec-V2` and `VLM2Vec` on representative examples spanning both video and document tasks, as detailed in Section A.6.

Figures 3 and 4 show that `VLM2Vec` mainly relies on surface-level visual or keyword similarity rather than the details of the whole video and the given text. In Figure 3, although the query asks about the vase that gets smashed, `VLM2Vec` only detects the existence of "a vase" and retrieves visually similar vases from unrelated scenes, without understanding which vase is actually involved. In Figure 4, it ignores the contextual and temporal cues in the video, retrieving frames that simply include the mentioned characters and a door, even though they are irrelevant to the described situation. In contrast, `VLM2Vec-V2` leverages the visual context from the entire video, allowing it to locate the correct moment that matches the intended meaning of the query. This indicates that `VLM2Vec-V2` learns a much stronger and more comprehensive video representation.

For document understanding, Figures 5 and 6 show that `VLM2Vec` mainly relies on keyword matching rather than reading and understanding the content of the whole document. In Figure 5, the incorrectly retrieved document contains the word "child", which appears in the query, but its actual content does not answer the question. Similarly, in Figure 6, `VLM2Vec` retrieves a document that mentions "social media", while ignoring the key details needed to provide the correct answer. In contrast, `VLM2Vec-V2` leverages the broader semantic information in the document and identifies the correct article that contains the detailed content required by the query. This demonstrates that `VLM2Vec-V2` offers a stronger capability in document comprehension beyond superficial keyword overlap.

### 4.5 Ablation Analysis

### 4.5.1 Generalization Across Modalities

To investigate how multimodal training configurations affect cross-modal generalization, we begin by examining the **single-modality training** results in Table 3. In the first column block, each model is trained on a single modality under a matched budget of 100K samples for a fair comparison. Under this controlled setting, models trained on image-only data consistently achieve the strongest cross-modal performance, demonstrating that high-quality image supervision serves as a crucial foundation for adapting MLLMs into unified multimodal embedding models. In contrast, models trained exclusively on VisDoc or Video data show substantial performance degradation across most evaluation modalities, especially on image-based benchmarks, indicating limited transferability when image supervision is absent.

Table 3 further analyzes the impact of **data source quality** within the second and third column blocks, where the modality and training scale remain fixed (100K samples) but the dataset source varies. Within VisDoc, `ColPali`-based training significantly outperforms both `VisRAG_{ind}` and `VisRAG_{syn}` across all modalities, suggesting that curated VisDoc datasets provide more effective supervision than weakly constructed or synthetic sources. A similar trend holds for video training: `Vid2Cap` yields clear advantages over `Cap2Vid` and `VideoQA`. These findings demonstrate that, even when the training modality is fixed, the dataset source critically dictates generalization capability.

| Eval. Modality | Training Modality | | | VisDoc Training Source | | | Video Training Source | | |
|---|---|---|---|---|---|---|---|---|---|
| | Image | VisDoc | Video | ColPali | $VisRAG_{ind}$ | $VisRAG_{syn}$ | Vid2Cap | Cap2Vid | VideoQA |
| Image | 58.45 | 39.87 | 30.64 | 32.29 | 4.14 | 3.53 | 20.48 | 3.52 | 4.12 |
| VisDoc | 58.73 | 61.72 | 41.55 | 53.66 | 1.26 | 0.79 | 34.23 | 0.80 | 0.56 |
| Video | 33.58 | 23.62 | 26.30 | 25.40 | 10.86 | 10.76 | 23.89 | 10.43 | 10.23 |
| AVG | 52.80 | 39.87 | 30.64 | 37.27 | 4.80 | 4.36 | 25.50 | 4.28 | 4.43 |

Table 3: Performance comparison across evaluation modalities (rows) under different training data settings (columns).Rows indicate the evaluation modality of the test datasets (Image, VisDoc, or Video). The first column block compares models trained with a single modality to analyze modality-level training effects. The second block varies VisDoc training data sources to study the quality of various VisDoc data sources. The third block varies Video training data sources to analyze the impact of video data sources. For fair comparison, each training configuration uses 100K training samples.

| Eval. Modality | Training Data Configuration | | | | |
|---|---|---|---|---|---|
| | Image 200K | Image 100K ColPali 100K | Image 100K VideoCap 100K | Image 100K ColPali 50K VideoCap 50K | All Sources 200K |
| Image | **60.75** | 59.01 | 58.23 | 59.20 | 58.62 |
| VisDoc | 56.69 | 64.45 | 56.33 | 63.40 | **66.87** |
| Video | 34.02 | 34.00 | 33.13 | **35.11** | 33.19 |
| AVG | 53.33 | 54.91 | 51.85 | 54.93 | **55.29** |

Table 4: Modality mixing analysis under different training data configurations. Rows correspond to evaluation modality, while columns represent different training data configurations formed by mixing modalities. For fair comparison, each configuration uses a fixed budget of 200K training samples.

We then analyze **multimodal training** using Table 4, which fixes a larger budget of 200K samples. Image-only training already provides a strong baseline across evaluation modalities due to the richness and transferability of image supervision. Building on this foundation, incorporating additional modalities can further improve performance, but the gains depend heavily on modality compatibility and data quality. Notably, mixing image data with low-quality video data may degrade performance, particularly for image- and video-based evaluations. In contrast, combining image supervision with high-quality VisDoc data consistently improves results across all modalities. Finally, the two full mixtures of Image + VisDoc + Video achieve the best overall average performance, demonstrating that broad modality coverage is beneficial for learning universal multimodal embeddings—provided that image supervision remains the core anchor and that additional modalities are carefully curated. To rigorously assess the generalization impact of multimodal training configurations, we further conduct a task-level paired $t$-test to evaluate statistical significance (Section A.5).

### 4.5.2 Ablation Study on Data Sampling Strategies

As part of our ablation study, we investigate the impact of *homogeneous sub-batching* on model performance across the three modalities. When the sub-batch size (IB) is set to 0, all samples in the batch are randomly drawn from different sources without grouping. A value of 64 indicates that a batch of size 1,024 is divided into sub-batches of size 64, resulting in data from 16 distinct sources per batch. At the other extreme, an IB of 1024 means the entire batch comes from a single source, effectively disabling interleaving. This setup allows us to analyze how different levels of source mixing influence training dynamics and cross-modal generalization.

As shown in Table 5, increasing the sub-batch size consistently improves performance for both VisDoc and Video. Conversely, the best performance on the Image modality is achieved with a sub-batch size of 64, exhibiting an inverted U-shaped trend—monotonically increasing from 0 to 64, followed by a monotonic decline from 64 to 1024.

| Modality | HSB-0 | HSB-32 | HSB-64 | HSB-128 | HSB-1024 |
|---|---|---|---|---|---|
| Image | 61.2 | 62.3 | **63.2** | 62.0 | 60.7 |
| VisDoc | 48.6 | 51.0 | 52.1 | 53.9 | **54.3** |
| Video | 34.6 | 33.2 | 33.5 | 34.5 | **35.4** |

Table 5: Performance comparison across different homogeneous sub-batch size for different modalities.

### 4.5.3 Ablation Study on Model Settings

We investigate the impact of different LoRA ranks (8, 16, 32) on model performance across modalities to understand how the capacity of parameter-efficient tuning affects generalization. As shown in the left part of Figure 2, a LoRA rank of 16 yields the best overall performance across image, video, and visual document tasks. This suggests that a moderate number of tunable parameters is beneficial for handling diverse modalities, while further increasing the rank to 32 does not lead to additional gains.

We also examine performance across training steps to understand how each modality benefits from continued training. As shown in the right part of Figure 2, all three modalities exhibit improved performance with increased training steps. Notably, there is no clear sign of saturation by 5K steps, particularly for VisDoc and Video, suggesting that further gains may be achievable with extended training. We leave a more in-depth exploration of long-horizon training and convergence behavior to future work.

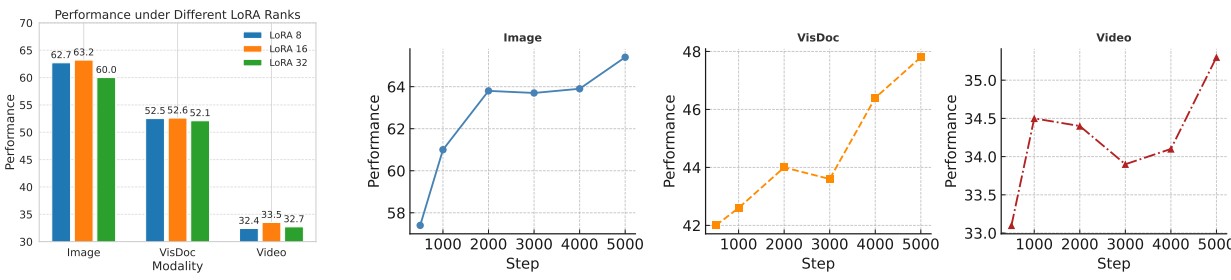

Figure 2: The left figure shows performance across LoRA ranks for different modalities, while the right figure illustrates performance trends across training steps.

## 5 Related Works

### 5.1 Multimodal Embedding Benchmarks

Numerous benchmarks have been proposed to evaluate multimodal models, with most early efforts focusing on static image-text pairs. Datasets such as MSCOCO (Lin et al., 2014), Flickr30K (Plummer et al., 2015), and Conceptual Captions (Sharma et al., 2018) enabled progress in tasks like image captioning and retrieval. Linear probing is also a common evaluation setting that trains a linear layer for image classification (Radford et al., 2021) to investigate the generalization of representation vectors. More recent benchmarks like M-BEIR (Wei et al., 2024) and MMEB (Jiang et al.) introduced multi-task evaluations for multimodal embedding models, covering tasks such as retrieval and QA. However, these benchmarks remain limited to static images and short contexts. Video-based benchmarks such as MSR-VTT (Xu et al., 2016), QVHighlights (Lei et al., 2021), and ActivityNet Captions (Krishna et al., 2017) target retrieval and captioning tasks, yet lack unified evaluation frameworks for embeddings. Our `MMEB-V2` addresses these gaps by providing a comprehensive embedding benchmark for diverse modalities.

### 5.2 Video Representation Learning

Video representation learning has evolved significantly, progressing from early convolutional approaches to sophisticated transformer-based architectures. Traditional vision-language models like CLIP (Radford

et al., 2021) and BLIP (Li et al., 2022), while effective for image-text tasks, often struggle to capture the temporal dynamics inherent in video data. To address this, recent models have been developed to better handle the complexities of video understanding. VideoCLIP (Xu et al., 2021), VideoCoCa (Yan et al., 2022) integrates contrastive learning with captioning objectives to enhance video-text representation alignment. InternVideo2 (Wang et al., 2024c) employs a progressive training approach that unifies masked video modeling, cross-modal contrastive learning, and next-token prediction, resulting in superior performance on over 60 video and audio tasks. Recent models like LLaVE (Lan et al., 2025) and LamRA (Liu et al., 2025), though trained exclusively on image-text data, have demonstrated the ability to generalize to text-video retrieval tasks in a zero-shot manner. These advancements highlight the ongoing efforts to develop models capable of effectively understanding and representing the complex temporal and semantic information in video data.

## 5.3 Visual Document Representation Learning

Visual document representation learning has become increasingly vital for tasks such as document retrieval and retrieval-augmented generation. Traditional text-based models often struggle to capture the rich visual and structural information present in documents, necessitating approaches that integrate both visual and textual modalities. One notable advancement is ColPali (Faysse et al.), which leverages vision-language models to enhance document retrieval efficiency by effectively capturing both textual and visual features. In the realm of retrieval-augmented generation, VisRAG (Yu et al., 2024) establishes a vision-based RAG pipeline that directly embeds documents as images using vision-language models, thereby preserving the original document information and outperforming traditional text-based RAG systems. Similarly, ViDoRAG (Wang et al., 2025) introduces a multi-agent framework tailored for complex reasoning across visual documents, employing a dynamic iterative reasoning process to enhance retrieval and generation tasks. Furthermore, benchmarks like MMLongBench-Doc (Ma et al., 2024) have been developed to assess long-context document understanding with visualizations, providing a comprehensive evaluation framework for multimodal models.

## 5.4 Unified Modality Representation Learning

Unified modality representation learning aims to build a single model capable of processing information across diverse data types, such as text, images, audio, and video. Some methods, like Uni-Retrieval (Jia et al., 2025), leverage multimodal large language models (MLLMs) and prompt-tuning to accommodate a variety of queries, achieving strong performance on universal benchmarks. While other recent studies have achieved remarkable results on diverse text and image tasks (Zhang et al., 2024; Thirukovalluru et al., 2025; Xiao et al., 2025; Jiang et al.), these models were not designed to unify image, video, and visual document retrieval within a single framework. Our latest work, VLM2Vec-V2, is the first to address this specific challenge. Following its release, subsequent efforts such as TTE (Cui et al., 2025) and Seed1.6-Embedding (Seed-Embedding-team) have proposed advanced techniques and demonstrated strong performance on our MMEB-V2 benchmark.

# 6 Conclusion

We introduced MMEB-V2, a comprehensive benchmark for evaluating multimodal embedding models across text, image, video, and visual document modalities. Alongside it, we proposed VLM2Vec-V2, a strong baseline trained via contrastive learning across a diverse range of tasks and modality combinations. Our extensive experiments demonstrate the effectiveness of VLM2Vec-V2 and the diagnostic value of MMEB-V2.

# 7 Broader Impact and Limitations

Multimodal embedding models are increasingly used in real-world systems such as document retrieval, multi-modal search, and content recommendation, where unified representations across heterogeneous modalities enable more accurate and efficient retrieval of relevant content. While our framework advances multimodal embedding learning, the MMEB-V2 benchmark is primarily constructed from English-language data and therefore may not adequately evaluate multilingual or cross-lingual capabilities. In addition, MMEB-V2 focuses on visual modalities and does not cover other important modalities such as audio, limiting its ability to assess truly

universal multimodal representations. We view `MMEB-V2` as a first step toward comprehensive multimodal evaluation and encourage future work to extend its coverage to additional languages and modalities.

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

# A Appendix

## A.1 Details of Baseline Models

**VLM2Vec** (Jiang et al.) converts vision-language models (VLMs) into the embedding models capable of handling diverse tasks. It reformulates all tasks as instruction-following ranking problems. Using contrastive learning and task-specific instructions, VLM2VEC learns to produce fixed-dimensional embeddings aligned across modalities.

**ColPali** (Faysse et al.) leverages a vision-language model trained to generate high-quality multi-vector embeddings from document page images. Combined with a late interaction matching mechanism, it achieves strong performance on visual document retrieval tasks.

**LamRA** (Liu et al., 2025) explores the use of large multimodal models (LMMs) for retrieval, unifying diverse retrieval tasks under a single framework without task-specific fine-tuning. It achieves this by employing two-stage training—language-only pretraining followed by multimodal instruction tuning—to enhance retrieval effectiveness.

**GME** (Zhang et al., 2024) is a unified multimodal embedding model finetuned from Qwen2-VL. It supports retrieval across single-modal, cross-modal, and fused-modal settings. GME is trained via contrastive learning using a diverse set of multimodal pairs including text, images, and image-text combinations.

## A.2 Details of Benchmark Construction

### A.2.1 Video Retrieval

**MSR-VTT** (Xu et al., 2016) is a dataset composed of 10K open-domain videos, each video clip ranging from 10 to 32 seconds in length and accompanied by a total of 200K captions. Following JSFusion (Yu et al., 2018), we sampled 1K clip-text pairs to incorporate into our benchmark. The query side contains both the instruction and the video caption, while the candidates consist of all 1K videos.

**DiDeMo** (Anne Hendricks et al., 2017) consists of 10K videos collected from Flickr, each trimmed to a maximum of 30 seconds. Each video includes approximately 3 to 5 annotated pairs of descriptions and their corresponding distinct moments. Following previous work (Liu et al., 2019; Luo et al., 2021), we concatenate these descriptions and perform "paragraph-to-video" retrieval on this benchmark. The official test split, which contains 1,004 paragraph-video pairs, is used.

**MSVD** (Chen & Dolan, 2011) contains 80K English descriptions for 1,970 YouTube videos, each ranging from 1 to 62 seconds in length. Each video is annotated with approximately 40 sentences. We use the official test split, which includes 670 videos, and select one sentence per video to construct 670 test cases.

**YouCook2** (Zhou et al., 2018) consists of 14K video clips sourced from 2K instructional cooking videos on YouTube. Each video contains multiple actions performed by the chef, accompanied by corresponding textual descriptions and temporal annotations. Each video clip is extracted and annotated with a single sentence. We follow the common practice (Miech et al., 2019) of using the validation split and removing videos that also appear in HowTo100M. Different papers may report slightly varying numbers of test cases, typically ranging from 3.1K to 3.3K. Our benchmark includes 3,179 clip-text pairs from YouCook2.

**VATEX** (Wang et al., 2019) contains 41,250 video clips sourced from Kinetics-600 dataset and 825K sentence-level descriptions. The public test set originally contained 6K videos. However, since many of them have been removed or set to private and are no longer accessible online, we use only a subset of 4,468 available videos. For each video, we select one description to include in our benchmark.

### A.2.2 Moment Retrieval

**QVHighlights** (Lei et al., 2021) is a dataset comprising 10K videos collected from YouTube, covering a diverse range of topics. Each video is annotated with high-quality labels for both query-based video moment retrieval and highlight detection. In our embedding benchmark, we adopt the standard practice of ranking

candidate clips and evaluating performance using Recall@1. In contrast, the QVHighlights paper and some other Vision-Language Models like InternVideo2 (Wang et al., 2024c) evaluate models using Recall@0.5 and Recall@0.7 with Intersection over Union (IoU) as a threshold, a metric that is not well-suited for embedding-based approaches.

**Charades-STA** (Gao et al., 2017), derived from the Charades (Sigurdsson et al., 2016) dataset, includes sentence-level temporal annotations for approximately 10K videos. Unlike its predecessor, Charades, Charades-STA replaces annotated action types with temporal sentences that describe actions. To minimize ambiguity in candidate clips, we created a filtered subset of the Charades-STA test set by applying a condition that selects videos where the relevant segment occupies less than one-third of the total video length.

**MomentSeeker** (Yuan et al., 2025) is a dataset designed to benchmark multimodal retrievers on long video moment retrieval tasks. Containing 1.6K queries, MomentSeeker consists of 4 subtasks with various query-side modalities. Additionally, MomentSeeker spans a diverse range of topics, including egocentric videos, cartoons, sports, and movies. For each query, we uniformly sampled nine negative clips and included all the ground truth clips as positive examples.

### A.2.3 Video Classification

**Kinetics-700-2020** (Carreira et al., 2022) is made up of approximately 648K Youtube video clips, covering a wide range of human actions, around 700 labels in total, such as cooking, driving, and drawing. Each video clip lasts 3 seconds on average. We sampled 1K video answer pairs from the validation set into our benchmark. The candidate texts are the list of all the labels. The raw video data are retrieved from CVD Foundation Github.

**Something Something v2** (Goyal et al., 2017) is the updated version of the Something Something v1 dataset. It consists of 220K crowd-source videos focusing on the physical interactions between humans and objects, with an average length of 4.03 seconds and a total of 174 action classes. We randomly sampled 1000 videos-text pairs from the validation split into our benchmark. The candidate texts are the list of all action classes. The raw video data are retrieved from Qualcomm.

**HMDB51** (Kuehne et al., 2011) is composed of 6K video clips, including both movies and web videos, with 51 action labels, such as catch, drink, and kick. We sampled 1K frames-text pairs from the test splits into our benchmark. The candidate texts are all labels. The raw video data are retrieved from the official website.

**Breakfast** (Kuehne et al., 2014) contains around 1.9K crowdsource video clips in the wild, more than 70 hours of total length, which are about preparing for 10 different types of breakfast, such as cereal, milk, pancakes, and fried eggs. There are 6 different camera viewpoints, and we only selected the clips filmed with camera 01, ignoring those filmed by other cameras. We used all the video clips of camera 01, around 433 samples in total. The candidate texts are the 10 types of breakfast. The raw video data are retrieved from the official website.

**UCF101** (Soomro et al., 2012) is an open domain video data set consisting of approximately 13K videos with 101 action categories, such as applying makeup, sports and playing instruments. We sampled 1K clip-text pairs from test splits into our benchmarks, and the candidate texts are all the action categories. The raw video data are retrieved from the official website.

### A.2.4 Video QA

While embedding models are not primarily designed for open-ended visual question answering, QA tasks offer a valuable way to assess whether a model can effectively understand visual inputs for different downstream purposes. They also enable fair comparison between embedding-based and generation-based approaches. To this end, we select multi-choice QA benchmarks that span a wide range of task types and are relatively less dependent on knowledge or reasoning abilities. We retain the original dataset configurations to ensure compatibility with prior work and facilitate direct comparison with existing models.

**MVBench** (Li et al., 2024) is a comprehensive benchmark designed to evaluate multi-modal large language models on video understanding, with a particular focus on temporal understanding. It defines 20 video tasks

covering a wide spectrum of temporal abilities – from perception to cognition – by transforming static tasks into dynamic, multiple-choice QA formats.

**Video-MME** (Fu et al., 2025) is a full-spectrum benchmark for evaluating Multi-modal Large Language Models (MLLMs) on video understanding. It consists of 2,700 manually annotated QA pairs based on 900 videos (totaling 254 hours). It ensures broad scenario coverage and captures diverse temporal dynamics by including a variety of video types and durations.

**NExT-QA** (Xiao et al., 2021) is a video question answering benchmark focused on causal and temporal reasoning in untrimmed daily activity videos. It supports both multiple-choice (what we use) and open-ended QA formats. In our study, we utilize only the multiple-choice portion to evaluate models' ability to reason about complex action dynamics and object interactions.

**EgoSchema** (Mangalam et al., 2023) is a diagnostic benchmark for long-form video understanding, constructed from Ego4D and comprising over 5,000 multiple-choice QA pairs spanning more than 250 hours of egocentric video. Each question is grounded in a 3-minute clip and targets long-range temporal reasoning. In our study, we use a subset of 500 questions for which answer annotations are publicly available.

### A.2.5   Visual Document Retrieval

**ViDoRe** (Faysse et al.; Macé et al., 2025) is a benchmark designed to evaluate document retrieval systems. The first version (v1) includes 10 subtasks. The dataset originates from two sources: (1) for academic tasks, it repurposes widely used visual question-answering (VQA) benchmarks, treating each question as a query and the corresponding page as the gold document; (2) for practical tasks, publicly accessible PDF documents are collected, and queries relevant to document pages are generated using Claude-3 Sonnet.

To address the saturation of the original ViDoRe benchmark, ViDoRe-v2 introduces more realistic and challenging retrieval tasks, including four new diverse and multilingual datasets.

**VisRAG** (Yu et al., 2024) serves as the test set for the VisRAG pipeline, which assesses multimodal retrievers in document retrieval. This benchmark consists of six subtasks adapted from VQA datasets, with a filtering process applied to exclude context-dependent queries unsuitable for retrieval.

**ViDoSeek** (Wang et al., 2025) is a large-scale document collection question-answering dataset originally designed to evaluate retrieval-augmented generation (RAG) performance requiring complex reasoning. We adapt it for retrieval by using questions as queries and reference pages as gold images, with each query linked to relevant images from a collection of approximately 5,000 images. The dataset covers diverse content types, including text, charts, tables, and structured layouts.

**MMLongBench-Doc** (Ma et al., 2024) is a long-context, multimodal VQA benchmark containing 1,082 expert-annotated questions. Unlike previous VQA datasets, it is built on 135 lengthy PDF documents, averaging 47.5 pages each. To ensure comprehensive evaluation, questions require evidence from multiple sources (text, images, charts, tables, and layout structures) and locations (e.g., specific page numbers). We repurpose this dataset for retrieval, treating questions as queries and evidence pages as gold images, with each query linked to relevant images from a collection of approximately 6,000 images.

### A.3   Detailed Scores

| | ColPali v1.3 | GME-2B | GME-7B | LamRA-Qwen2 | LamRA-Qwen2.5 | VLM2Vec-2B | VLM2Vec-7B | VLM2Vec-V2.0 |
|---|---|---|---|---|---|---|---|---|
| Avg - All (78 tasks) | 46.0 | 54.1 | 57.8 | 40.4 | 47.4 | 47.0 | 52.3 | 58.0 |
| Avg - Image (36 tasks, Hit@1) | 34.9 | 51.9 | 56.0 | 54.1 | 52.4 | 59.7 | 65.5 | 64.9 |
| Avg - Video (18 tasks, Hit@1) | 28.2 | 33.6 | 38.4 | 35.0 | 33.6 | 28.6 | 33.7 | 34.6 |
| Avg - Visdoc (24 tasks, NDCG@5) | 75.8 | 72.7 | 75.2 | 23.9 | 50.2 | 41.6 | 46.4 | 65.4 |
| I-CLS (10) | 40.3 | 54.4 | 57.7 | 59.2 | 51.7 | 58.7 | 62.7 | 62.9 |
| I-QA (10) | 11.5 | 29.9 | 34.7 | 26.5 | 34.1 | 49.3 | 56.9 | 56.3 |
| I-RET (12) | 48.1 | 66.9 | 71.2 | 70.0 | 66.9 | 65.0 | 69.4 | 69.5 |
| I-VG (4) | 40.3 | 55.5 | 59.3 | 62.7 | 56.7 | 72.9 | 82.2 | 77.3 |
| V-CLS (5) | 26.7 | 34.9 | 37.4 | 39.3 | 32.9 | 33.4 | 39.1 | 39.3 |
| V-QA (5) | 37.8 | 42.0 | 50.4 | 42.6 | 42.6 | 30.5 | 30.0 | 34.3 |
| V-RET (5) | 21.6 | 25.6 | 28.4 | 24.3 | 23.2 | 20.6 | 29.0 | 28.8 |
| V-MR (3) | 25.5 | 31.1 | 37.0 | 32.8 | 37.2 | 30.7 | 38.9 | 36.8 |
| VD-Vidore-V1 (10) | 83.6 | 86.1 | 89.4 | 22.0 | 56.3 | 49.8 | 56.9 | 75.5 |
| VD-Vidore-V2 (4) | 52.0 | 54.0 | 55.6 | 11.5 | 33.3 | 13.5 | 9.4 | 44.9 |
| VD-VisRAG (6) | 81.1 | 82.5 | 85.0 | 37.4 | 58.2 | 51.8 | 59.1 | 79.4 |
| VD-OOD (4) | 43.1 | 43.1 | 44.4 | 21.0 | 40.1 | 33.5 | 38.1 | 39.4 |
| ImageNet-1K | 42.4 | 58.3 | 64.6 | 72.3 | 58.9 | 77.5 | 80.1 | 80.8 |
| N24News | 25.5 | 50.1 | 50.5 | 51.3 | 29.8 | 73.7 | 79.7 | 72.9 |
| HatefulMemes | 50.6 | 52.5 | 53.6 | 49.0 | 51.3 | 58.3 | 69.7 | 56.3 |
| VOC2007 | 69.8 | 75.9 | 80.3 | 80.1 | 78.7 | 74.3 | 80.7 | 85.0 |
| SUN397 | 56.1 | 67.3 | 69.5 | 68.5 | 66.5 | 73.8 | 77.4 | 71.0 |
| Place365 | 27.5 | 35.8 | 39.1 | 40.6 | 37.4 | 35.3 | 37.4 | 35.9 |
| ImageNet-A | 14.9 | 28.8 | 41.2 | 47.0 | 36.3 | 50.9 | 58.1 | 47.4 |
| ImageNet-R | 64.6 | 78.6 | 83.9 | 88.5 | 77.0 | 84.7 | 73.9 | 89.3 |
| ObjectNet | 45.6 | 70.6 | 69.0 | 66.4 | 59.4 | 37.1 | 40.1 | 65.2 |
| Country211 | 6.0 | 26.5 | 24.8 | 28.3 | 21.7 | 21.5 | 29.8 | 25.2 |
| OK-VQA | 9.4 | 29.9 | 33.2 | 37.8 | 39.9 | 48.5 | 56.8 | 51.5 |
| A-OKVQA | 6.6 | 18.6 | 21.0 | 27.0 | 34.1 | 39.5 | 47.3 | 43.6 |
| DocVQA | 11.3 | 29.8 | 41.4 | 22.3 | 37.1 | 82.5 | 89.7 | 90.1 |
| InfographicsVQA | 5.0 | 11.6 | 20.3 | 16.5 | 23.7 | 47.7 | 60.0 | 58.8 |
| ChartQA | 5.7 | 13.4 | 17.8 | 11.7 | 15.0 | 42.3 | 56.9 | 47.4 |
| Visual7W | 6.1 | 16.2 | 22.2 | 19.6 | 24.6 | 51.2 | 52.7 | 52.9 |
| ScienceQA | 16.3 | 27.3 | 28.0 | 26.3 | 31.3 | 30.7 | 38.5 | 38.2 |
| VizWiz | 27.6 | 37.0 | 39.0 | 32.0 | 32.0 | 38.6 | 39.9 | 43.3 |
| GQA | 8.3 | 75.1 | 76.9 | 38.5 | 57.4 | 48.3 | 55.1 | 64.9 |
| TextVQA | 18.8 | 39.7 | 46.8 | 33.0 | 46.1 | 63.3 | 71.6 | 72.2 |
| VisDial | 41.2 | 48.1 | 60.8 | 61.3 | 62.5 | 74.3 | 81.9 | 82.7 |
| CIRR | 8.2 | 44.2 | 54.9 | 51.7 | 44.7 | 46.8 | 51.1 | 57.5 |
| VisualNews_t2i | 50.1 | 74.7 | 79.7 | 70.4 | 70.1 | 73.1 | 80.5 | 74.5 |
| VisualNews_i2t | 47.6 | 78.3 | 83.6 | 83.9 | 74.2 | 73.7 | 81.2 | 78.2 |
| MSCOCO_t2i | 59.2 | 68.1 | 71.2 | 72.2 | 65.7 | 73.4 | 77.2 | 75.3 |
| MSCOCO_i2t | 49.9 | 63.1 | 57.7 | 73.7 | 71.1 | 68.5 | 73.9 | 71.4 |
| NIGHTS | 65.5 | 67.0 | 67.6 | 65.6 | 64.4 | 66.3 | 67.6 | 68.6 |
| WebQA | 53.8 | 88.8 | 91.4 | 81.0 | 85.7 | 85.9 | 88.3 | 90.6 |
| FashionIQ | 5.9 | 32.9 | 37.8 | 42.0 | 33.4 | 14.0 | 17.1 | 19.5 |
| Wiki-SS-NQ | 80.5 | 73.9 | 78.2 | 69.7 | 67.0 | 54.2 | 62.3 | 66.9 |
| OVEN | 50.0 | 72.3 | 75.1 | 82.0 | 84.8 | 68.3 | 66.5 | 64.3 |
| EDIS | 64.7 | 91.8 | 96.0 | 85.9 | 78.7 | 81.2 | 85.7 | 84.1 |
| MSCOCO | 36.7 | 28.6 | 31.4 | 44.8 | 36.0 | 66.5 | 75.7 | 67.1 |
| RefCOCO | 64.5 | 55.9 | 60.9 | 62.8 | 57.1 | 80.9 | 87.6 | 87.1 |
| RefCOCO-Matching | 3.9 | 73.3 | 78.4 | 75.7 | 82.6 | 75.7 | 84.6 | 85.8 |
| Visual7W-Pointing | 56.1 | 64.1 | 66.5 | 67.3 | 51.2 | 68.3 | 81.0 | 69.2 |
| K700 | 23.4 | 35.2 | 39.7 | 42.3 | 32.1 | 31.4 | 35.5 | 38.0 |
| SmthSmthV2 | 25.1 | 29.9 | 30.6 | 36.3 | 25.3 | 30.9 | 32.1 | 42.8 |
| HMDB51 | 24.8 | 43.4 | 47.9 | 40.5 | 33.8 | 33.8 | 42.2 | 40.9 |
| UCF101 | 49.4 | 52.4 | 54.7 | 60.4 | 53.0 | 57.5 | 61.8 | 60.0 |
| Breakfast | 10.9 | 13.6 | 14.3 | 16.9 | 20.1 | 13.4 | 23.8 | 14.8 |
| MVBench | 33.7 | 37.5 | 46.6 | 37.2 | 37.6 | 30.5 | 28.5 | 33.7 |
| Video-MME | 30.6 | 34.3 | 39.2 | 34.1 | 35.1 | 26.9 | 27.8 | 30.7 |
| NExTQA | 35.2 | 39.5 | 53.6 | 43.7 | 44.9 | 20.3 | 20.3 | 20.9 |
| EgoSchema | 38.4 | 40.8 | 46.8 | 44.8 | 47.0 | 25.4 | 21.8 | 34.0 |
| ActivityNetQA | 51.3 | 58.0 | 65.6 | 53.2 | 48.5 | 49.6 | 51.4 | 52.3 |
| DiDeMo | 22.8 | 22.0 | 26.4 | 24.8 | 22.8 | 19.4 | 29.3 | 30.4 |
| MSR-VTT | 17.6 | 27.3 | 31.8 | 22.1 | 25.0 | 25.2 | 34.5 | 28.3 |
| MSVD | 45.4 | 47.6 | 49.7 | 46.1 | 41.9 | 38.2 | 46.7 | 48.1 |
| VATEX | 16.7 | 23.0 | 24.9 | 19.1 | 18.7 | 16.2 | 25.5 | 26.5 |
| YouCook2 | 5.3 | 7.9 | 9.1 | 9.2 | 7.5 | 4.1 | 9.0 | 10.6 |
| QVHighlight | 19.9 | 43.6 | 59.5 | 53.8 | 60.9 | 44.2 | 57.7 | 49.4 |
| Charades-STA | 29.0 | 14.9 | 14.0 | 10.9 | 18.8 | 13.6 | 19.8 | 20.2 |
| MomentSeeker | 29.3 | 37.1 | 39.3 | 35.9 | 33.3 | 35.4 | 41.0 | 42.9 |
| ViDoRe_arxivqa | 81.7 | 82.8 | 86.9 | 10.8 | 53.0 | 48.9 | 60.2 | 80.6 |
| ViDoRe_docvqa | 56.6 | 53.1 | 57.5 | 19.1 | 25.4 | 27.0 | 34.7 | 44.9 |
| ViDoRe_infovqa | 84.9 | 90.2 | 91.6 | 46.3 | 72.3 | 67.2 | 70.4 | 83.7 |
| ViDoRe_tabfquad | 86.9 | 93.3 | 94.6 | 42.8 | 66.1 | 62.6 | 78.2 | 89.2 |
| ViDoRe_tatdqa | 70.9 | 69.9 | 74.1 | 11.4 | 25.9 | 19.8 | 27.6 | 43.8 |
| ViDoRe_shiftproject | 75.1 | 89.5 | 96.8 | 12.0 | 27.3 | 41.8 | 38.6 | 60.8 |
| ViDoRe_artificial_intelligence | 95.7 | 97.5 | 99.6 | 10.3 | 72.0 | 55.0 | 67.7 | 88.5 |
| ViDoRe_energy | 94.7 | 91.9 | 95.3 | 24.8 | 65.2 | 59.1 | 60.4 | 86.5 |
| ViDoRe_government_reports | 93.6 | 94.6 | 98.8 | 16.4 | 72.2 | 57.1 | 61.8 | 85.0 |
| ViDoRe_healthcare_industry | 95.9 | 98.7 | 99.3 | 25.9 | 83.8 | 59.6 | 69.9 | 92.2 |
| ViDoRe_esg_reports_human_labeled_v2 | 51.3 | 61.0 | 63.4 | 7.6 | 33.0 | 12.6 | 6.8 | 45.6 |
| ViDoRe_biomedical_lectures_v2_multilingual | 54.7 | 54.0 | 49.5 | 13.3 | 35.9 | 7.4 | 5.1 | 44.3 |
| ViDoRe_economics_reports_v2_multilingual | 49.0 | 50.2 | 54.2 | 19.1 | 31.9 | 13.9 | 13.9 | 43.0 |
| ViDoRe_esg_reports_v2_multilingual | 52.9 | 50.7 | 55.4 | 5.9 | 32.5 | 20.1 | 11.9 | 46.6 |
| VisRAG_ArxivQA | 80.9 | 82.0 | 87.4 | 2.0 | 37.7 | 41.8 | 52.6 | 76.9 |
| VisRAG_ChartQA | 78.2 | 79.9 | 81.9 | 41.3 | 65.9 | 57.9 | 70.2 | 84.4 |
| VisRAG_MP-DocVQA | 86.8 | 84.4 | 89.2 | 33.4 | 54.5 | 43.2 | 52.8 | 71.8 |
| VisRAG_SlideVQA | 95.0 | 93.4 | 94.5 | 56.5 | 76.5 | 74.0 | 72.8 | 91.5 |
| VisRAG_InfoVQA | 85.7 | 91.4 | 93.5 | 56.3 | 73.3 | 70.7 | 72.0 | 85.7 |
| VisRAG_PlotQA | 60.3 | 64.1 | 63.4 | 34.6 | 41.2 | 23.4 | 34.4 | 66.1 |
| ViDoSeek-page | 22.2 | 21.6 | 23.2 | 11.3 | 23.1 | 17.7 | 22.3 | 21.9 |
| ViDoSeek-doc | 83.7 | 83.6 | 83.9 | 37.1 | 80.3 | 74.3 | 77.8 | 80.2 |
| MMLongBench-page | 14.2 | 15.8 | 16.2 | 8.0 | 13.5 | 9.6 | 11.8 | 11.9 |
| MMLongBench-doc | 52.5 | 51.4 | 54.3 | 27.6 | 43.5 | 32.6 | 40.5 | 43.7 |

Table 6: Performance comparison of various models on the full `MMEB-V2` benchmark, covering 78 tasks across image, video, and visual document modalities. Numbers in parentheses represent the task count for each category.

| Rank | Model | Size (B) | Overall | Image | Video | VisDoc |
|------|-------|----------|---------|-------|-------|--------|
| 1 | seed1.6-embedding-1215 | – | 75.59 | 77.99 | 67.74 | 77.88 |
| 2 | WeMM-Embedding-8B | 8.77 | 74.42 | 78.09 | 63.24 | 77.30 |
| 3 | IFM-TTE-7B | 8.29 | 74.07 | 77.90 | 59.19 | 79.48 |
| 4 | WeMM-Embedding-2B | 2.13 | 71.67 | 76.08 | 58.67 | 74.81 |
| 5 | RzenEmbed-v2-7B | 8.29 | 71.61 | 75.92 | 55.73 | 77.06 |
| 6 | seed-1.6-embedding | – | 71.27 | 77.78 | 55.34 | 73.44 |
| 7 | RzenEmbed-v1-7B | 8.29 | 68.88 | 73.60 | 48.87 | 76.80 |
| 8 | Ops-MM-embedding-v1-7B | 8.29 | 67.61 | 72.72 | 53.76 | 70.34 |
| 9 | UME-R1-7B | 8.29 | 64.50 | 71.25 | 47.50 | 67.13 |
| 10 | RzenEmbed-v1-2B | 2.21 | 64.36 | 68.53 | 42.62 | 74.41 |
| 11 | Ops-MM-embedding-v1-2B | 2.21 | 63.44 | 69.03 | 47.56 | 66.96 |
| 12 | interestFM-UIR-CAFe-7B | 8.03 | 60.63 | 67.56 | 42.40 | 63.92 |
| 13 | UME-R1-2B | 2.21 | 60.11 | 66.56 | 42.23 | 63.86 |
| 14 | UniME-V2-LLaVA-OneVision-7B | 8.03 | 59.56 | 71.77 | 39.01 | 56.68 |
| 15 | VLM2Vec-V2.0-Qwen2VL-2B | 2.21 | 58.02 | 64.85 | 34.58 | 65.36 |

Table 7: Leaderboard results on the MMEB-V2 benchmark as of December 2025. Scores are reported for overall performance as well as for modality-specific subsets, including Image, Video, and VisDoc.

## A.4   Updated Results on the MMEB-V2 Benchmark

Since the release of the MMEB series benchmarks, they have become widely adopted as standard evaluations for MLLM embeddings. As of December 2025, more than 50 models have been evaluated on MMEB-V1 and over 30 models on MMEB-V2. In this appendix, we include additional results from several recent state-of-the-art multimodal embedding models evaluated on the MMEB-V2 benchmark.

| Training Data Configuration | Image Δ | | Video Δ | | VisDoc Δ | | Global Δ | |
|---|---|---|---|---|---|---|---|---|
| Image 200K (Baseline) | − | | − | | − | | − | |
| Image 100K + ColPali 100K | −1.74 | | −0.02 | | +7.76 | *** | +1.58 | * |
| Image 100K + VideoCap 100K | −2.52 | *** | −0.89 | | −0.36 | | −1.48 | *** |
| Image 100K + ColPali 50K + VideoCap 50K | −1.55 | ** | +1.09 | * | +6.69 | *** | +1.59 | ** |
| All Sources 200K | −2.13 | ** | −0.84 | | +10.19 | *** | +1.96 | * |

Table 8: Task-level significance evaluation of different modality mixing strategies compared to the Image 200K baseline. Listed values indicate the mean performance change across 78 MMEB-V2 tasks, with positive values reflecting consistent gains. Statistical reliability is assessed using paired $t$-tests: * $p < 0.05$, ** $p < 0.01$, *** $p < 0.001$.

### A.5 Significance Evaluation of Modality Mixing Strategies

To rigorously assess the generalization impact of multimodal training configurations (Section 4.5.1; Table 4), we conduct a task-level paired $t$-test to evaluate statistical significance. Instead of treating individual queries as independent observations—which would artificially inflate significance due to large sample counts—we treat each dataset (task) as one independent sample ($N = 78$ tasks). For each ablation model $M$, we compute a paired difference vector $D = [\text{Score}_{M,1} - \text{Score}_{\text{Base},1}, \ldots, \text{Score}_{M,N} - \text{Score}_{\text{Base},N}]$, comparing its performance to that of the Image-only baseline on a per-task basis. We then test the null hypothesis that the mean of $D$ is zero, where the sign of the mean difference reflects improvement or degradation, and the $p$-value indicates whether the effect is statistically reliable across the full task distribution.

The significance results in Table 8 confirm the key observations from Section 4.5.1. In particular, **effective modality mixing requires both high-quality data and modality compatibility**: adding curated VisDoc supervision (e.g., `ColPali`) leads to statistically significant gains, whereas incorporating noisier video supervision (`VideoCap`) can negatively affect performance. Furthermore, **the two full Image + VisDoc + Video mixtures deliver the strongest global improvements**, highlighting that broader modality coverage is most beneficial when additional modalities are carefully selected and curated.

### A.6 Example Cases

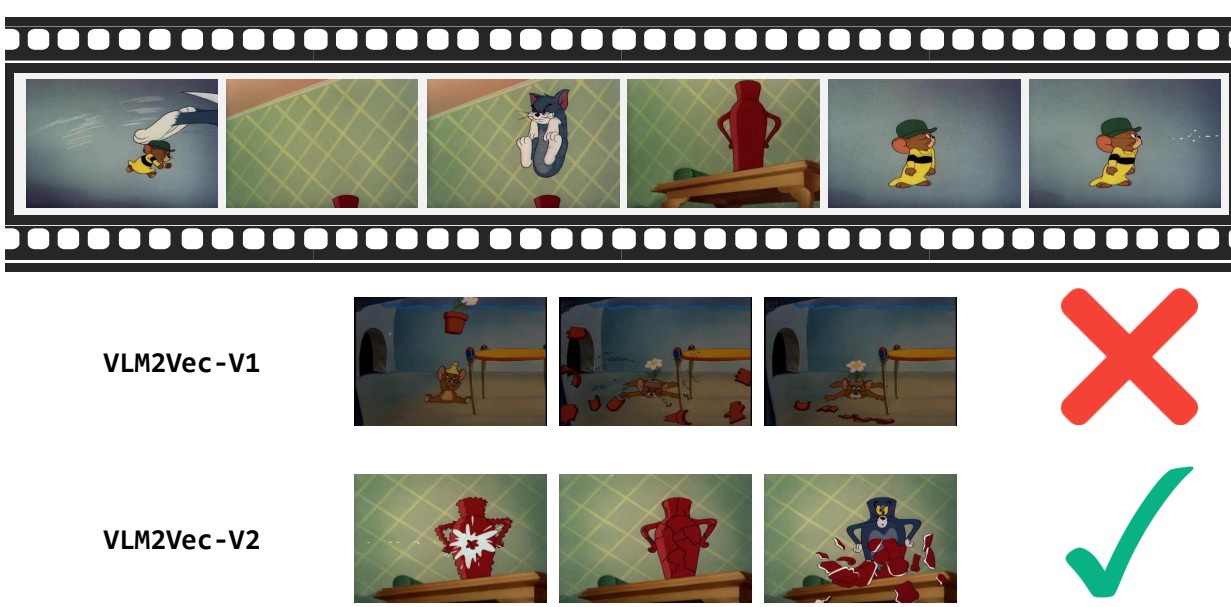

Figure 3: Example of improved video understanding in VLM2Vec-V2 over VLM2Vec.

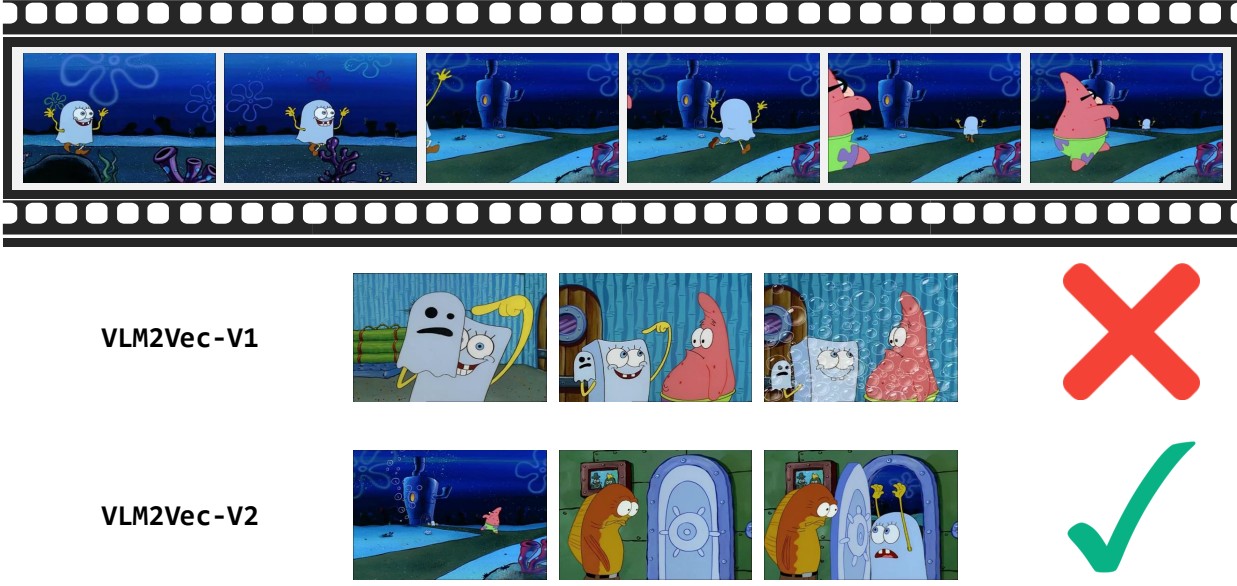

Figure 4: Example of improved video understanding in VLM2Vec-V2 over VLM2Vec.

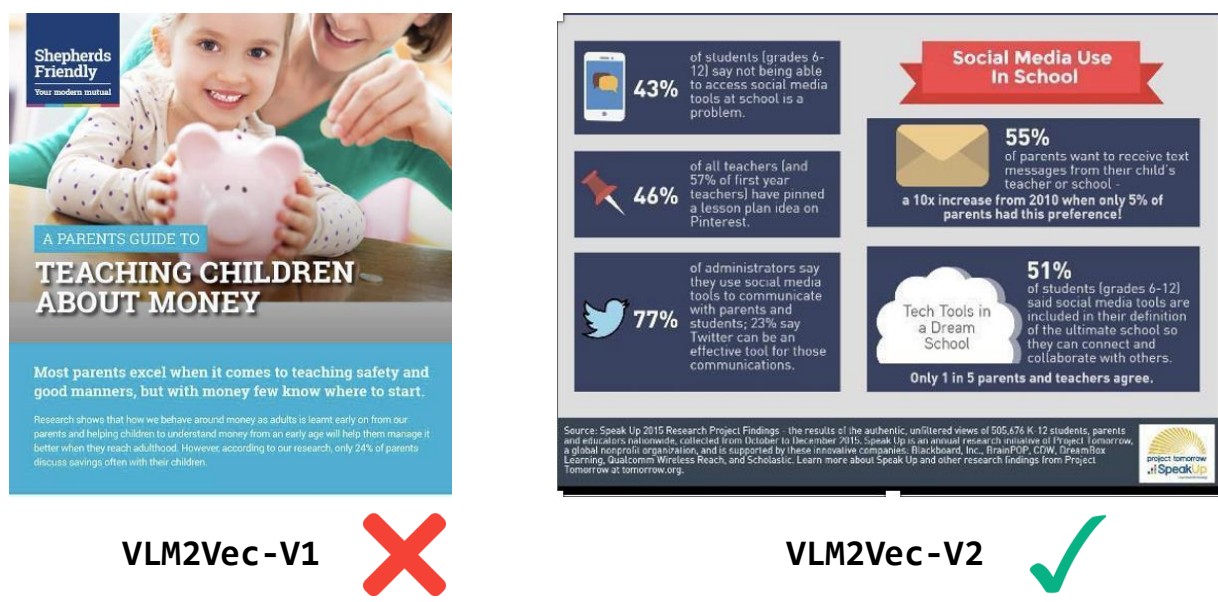

Figure 5: Example of improved document understanding in VLM2Vec-V2 over VLM2Vec. The images are cropped for clearer presentation.

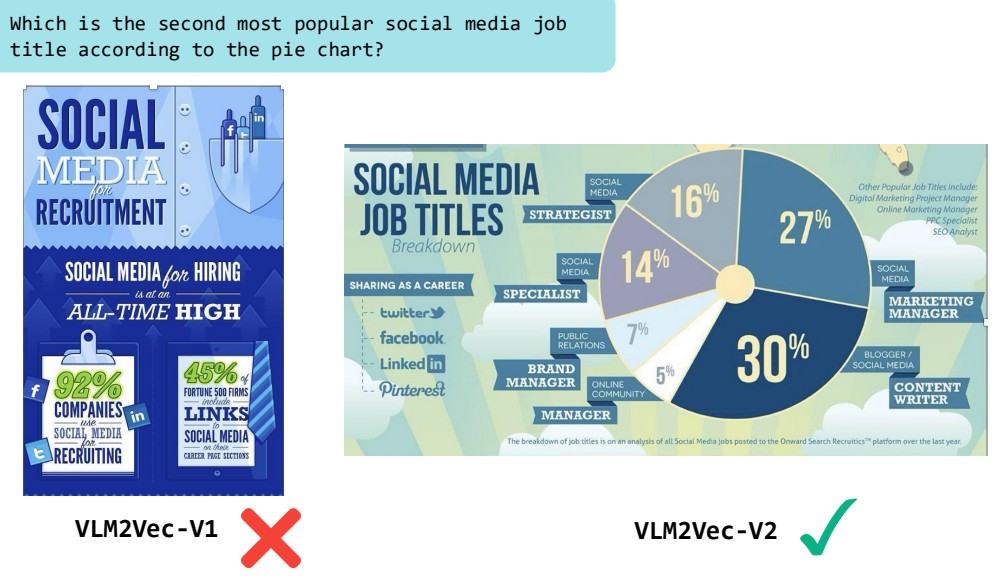

Figure 6: Example of improved document understanding in VLM2Vec-V2 over VLM2Vec. The images are cropped for clearer presentation.

