# OpenReview forum: "VLM2Vec-V2: Advancing Multimodal Embedding for Videos, Images, and Visual Documents"
_TMLR — Accepted by TMLR_

### Review · Reviewer_Lus6 · 2025-12-03

**Summary Of Contributions:**

This paper presents a new dataset (MMEB-V2) and model (VLM2VEC-V2) for multimodal embeddings. MMEB-V2 builds upon prior work by incorporating new datasets spanning five new tasks focusing on video and visual document understanding. The proposed model outperforms other relevant multimodal embedding baselines (of a similar size) across a wide variety of 78 tasks.

**Strengths**:
- The scope of datasets, tasks, modalities, etc. included is very impressive. This is an all-encompassing effort with diverse downstream applications.
- Experiments are very thorough, spanning many tasks of all varieties. Competitive, relevant baselines are chosen as well.
- The style of writing is generally straightforward and clear.

**Weaknesses**:
- This reads more like a technical report than a research paper necessarily. There are often instances of informal language and brief/cursory discussions of concepts.
- Unless I am missing this, there is no mention of code or data access. For this type of paper, details about code/data release will be very important to determine its utility.

**Audience:**

Yes

**Audience Explanation:**

This paper will surely be of interest to the TMLR readership, particularly if it is accompanied by the open-source release of code, data, and/or model weights. The idea of an openly accessible multimodal embedding model spanning so many different tasks and modalities will be highly appealing to certain readers.

**Claims And Evidence:**

Yes

**Claims Explanation:**

Experiments demonstrate the utility of the proposed model against relevant baseline multimodal embedding models across a wide variety of tasks.

**Requested Changes:**

**Main feedback**
- Ensure that each paragraph has a dedicated purpose and is fully fleshed out. If claims are made, they should always be substantiated by relevant citations when possible. E.g., is there a reference that can support a claim like the following? "The lack of coverage causes the existing embedding models to fall behind on many realistic tasks like article searching, website searching, Youtube video search, etc."
- Comment on the availability of code, data, and model weights.
- How do these general multimodal embedding models compare, e.g., to “specialized” models in individual tasks and domains? This would be helpful with regard to understand the utility of such a general approach.
- What is the unique value proposition of VLM2Vec-V2? I see that it outperforms other 2B-parameter models, but is on par with GME 7B, for example. Practically speaking, what is the difference from an end user’s perspective between using a 2B and 7B model? Make this value proposition very clear upfront, even though it is stated across several different places in the paper.

**Minor feedback**
- Abstract: “These datasets fail to cover broader forms of visual information like documents, pdf, websites, videos, slides, etc. The lack of coverage causes the existing embedding models to fall behind on many realistic tasks like article searching, website searching, Youtube video search, etc.” I would avoid repeating “etc.” multiple times; e.g., could write “tasks such as article searching…”
- Sec 3.2: “where tau is a temperature”. I would add “hyperparameter” to the end of this
- Sec 3.2: Example of informal tone: “we feed query and target into it”

---

> ### Author Response · Authors · 2025-12-24
> **Rebuttal by Authors**
>
> We sincerely appreciate your thoughtful comments and your recognition of the "impressive scope" and "thorough experiments" presented in our work. We have carefully addressed your feedback regarding informal language and unclear claims in the revised manuscript, where updates are now **highlighted in blue text**. These revisions include appropriate citations and specific empirical evidence to substantiate our findings as requested.
>
> In this paper, we aim to provide a comprehensive model training guideline specifically designed for an academic lab budget, utilizing a 2B-parameter model and publicly available training data. A critical contribution of this work is the MMEB benchmark , which we believe is the first of its kind to offer an all-encompassing evaluation for multimodal embeddings across diverse tasks and modalities. Since its introduction, MMEB has become one of the most widely adopted benchmarks in this domain; to date, more than 30 different models have reported their scores on our public leaderboard. We hope this highlights the unique value proposition of our work in fostering scalable and adaptable representation learning for both the research community and real-world applications.
>
>
> >Comment on the availability of code, data, and model weights.
>
> We appreciate the importance of accessibility for the utility of this work. All code, curated datasets, and model weights have already been released and are actively maintained by the authors. However, to strictly comply with the double-blind review policy, we have omitted these links from the current submission. We will include the full links to our GitHub repository and the public MMEB-V2 leaderboard in the final version of the paper.
>
>
> >General multimodal embedding models vs. specialized models
>
> While specialized models achieve high performance in targeted domains, they often fail to generalize. For example, a document-specialized model like ColPali achieves strong results on document tasks (71.0%) but drops severely on image (34.9%) and video (28.2%) tasks. A general model like VLM2Vec-V2 offers simplified deployment, handles diverse data types simultaneously, and provides a robust starting point for further fine-tuning.
>
>
> >What is the unique value proposition of VLM2Vec-V2, and why should an end user prefer a 2B-parameter model over larger 7B-parameter alternatives?
>
> VLM2Vec-V2 (2B) achieves an overall score of 58.0, outperforming larger baselines like GME-7B (57.8). Our focus on the 2B scale is a deliberate choice to provide a training recipe under an academic lab budget, based on the following practical advantages:
> * **Training Efficiency**: Multimodal training with visual documents and videos is computationally intensive due to long context lengths (up to 4k tokens). The 2B backbone allowed us to explore training recipes and hyperparameter tuning that would be cost-prohibitive with 7B+ models.
> * **Lower Inference Latency**: The 2B model naturally offers faster processing and lower inference costs for real-world applications.
> * **Reduced Serving and Storage Costs**: A critical advantage lies in the embedding vector size. The hidden state (embedding) size for Qwen2-VL 2B is 1536, compared to 3584 for the 7B version. This 57% reduction in dimensionality means:
>    * **Smaller Index Size**: Storing billions of embeddings requires significantly less memory and disk space.
>    * **Faster Retrieval**: Vector similarity searches (e.g., in a vector database) are much faster with smaller dimensions, directly improving the scalability of RAG and search systems.
>
>
> >Feedback related to the paper writing
>
> We have updated the manuscript in response to both your valuable suggestions regarding the claims in the introduction and the points raised in the minor feedback section.

---

### Review · Reviewer_5ybX · 2025-12-03

**Summary Of Contributions:**

The authors provide two contributions to the development of multimodal models.
1. They builds a new datasets named MMEB-V2, aiming to provide an extensive benchmark for a variety of multimodal tasks. This new dataset is not obtained from the creation of new instances, but by extending the MMEB benchmark with tasks borrowed from other datasets in the literature. Still, a consequent work has been done to unify all datasets.
2. They provide a new multimodal model, named  VLM2Vec-V2, that provides a unified treatment for learning embedding intended to be used to solve downstream multimodal tasks (including videos, images and documents). They compared the accuracy of this new model to several baselines on the proposed MMEB-V2 benchmark, resulting in an extensive and convincing study.

**Additional Comments:**

If I understand correctly, the datasets included in the MMEB-v2 benchmark are completely distinct from the ones used to train the embedding model VLM2Vec-V2. How did the authors decide which dataset to include in the benchmark and which ones to include in the training dataset? Are we confident that the baseline approaches have not been trained on datasets included in the benchmark?

**Audience:**

Yes

**Audience Explanation:**

I am not certain of my assessment here, but I guess researchers working on Large Models and Multimodalities would benefit from the proposed extended dataset (MMEB-V2) and a baseline for comparing future approaches (VLN2Vec-V2)

**Broader Impact Concerns:**

The manuscript does not contain any broader impact statement. As multimodal models are intended for deployment across a variety of real-world contexts, I think the work deserves a statement on their societal impact (positive and negative). A discussion of the potential shortcomings of the MMEB-V2 dataset's content would also be of interest.

**Claims And Evidence:**

Yes

**Claims Explanation:**

I would like first to state that I don't feel qualified to conduct this review, as I am not aware of the current state of Large Multimodal Models.

That said, I acknowledge that the empirical evaluation of VLM2Vec-V2 is extensive, and the results are convincing. I appreciate the presence of a proper ablation study.

One caveat is that the introduction announces more fundamental answers than the paper provides:
> Through VLM2Vec-V2 and MMEB-V2, we aim to investigate the following research questions: How well can a multimodal embedding generalize across diverse visual modalities? What are the key ingredients for training robust and versatile multimodal embedding models? What are the key challenges in representing temporal information in videos and structured information in visual documents?

The last two questions are not explicitly answered in the manuscript.

**Requested Changes:**

I consider all the following adjustments important:
1. It isn't easy to know what the difference is between the VLM2Vec-V2 model and previous models, especially the changes made to the first version (VLM2Vec). This information should be clearly stated. Related to this, I would like to assess the contributions at a more fine-grained level: which components of the model or training strategies are new?
2. The result tables provide "performance" metrics, but I could not find the precise metrics for each task.
3. Section 5.4 suggests that the proposed models and benchmarks have been released before submitting the paper to TMLR, and other models have been tested in the meantime (TTE and Seed1.6-Embedding). These results should be included in the manuscript.
4. More information on the resources needed to train the VLM2Vec-V2 model. Section 4.1.2 states that experiments were performed on 8 NVIDIA H100 GPUs. How long does it take to train VLM2Vec-V2? How does it compare to baseline models?
5. The bibliography needs to be proofread to include the references to published works (accepted after a peer-reviewed process) in place of their arXiv counterparts.
     - BehnamGhader et al. (2024) was accepted at COLM 2024
     - Faysse et al (2024) was accepted at ICLR 2025
     - Jia et al. (2025) was accepted at ACL 2025
     - and so on

---

> ### Author Response · Authors · 2025-12-24
> **Rebuttal by Authors (Part 1)**
>
> We sincerely appreciate your constructive feedback and your recognition of the "extensive and convincing study" presented in our empirical evaluation. We have addressed your specific concerns in the revised manuscript as follows:
>
> >Which components of the model or training strategies are new.
>
> We have reorganized **Section 3** to clearly differentiate our new contributions from the original framework. **Section 3.1** provides a preliminary overview of the VLM2Vec framework and **Section 3.2** describes the foundation model selection, while **Sections 3.3–3.4** detail the novel extensions in V2 . These include our unified multimodal data mixing strategy and the homogeneous sub-batching scheme designed to balance heterogeneous data sources and increase contrastive difficulty . Furthermore, we emphasize that the **MMEB-V2 benchmark** is a major contribution, providing a standardized protocol for evaluating and training next-generation multimodal embeddings.
>
>
> >Clearly specifying the exact evaluation metrics used for each task.
>
> As suggested, we have updated **Section 4.2** to explicitly define the metrics for each modality . We utilize **Hit@1** for all video and image tasks to measure top-rank accuracy. For visual document tasks, we report **NDCG@5** to ensure consistency with existing benchmarks in that domain.
>
>
> >Since the models and benchmarks were released prior to submission and newer models (e.g., TTE and Seed1.6-Embedding) have since been evaluated, their results should be incorporated into the paper.
>
> **Appendix A.4** has been updated to incorporate the most recent leaderboard results as of December 2025 . Since the release of the MMEB series, it has become a standard evaluation in the domain; as of late 2025, over **50 models** have been evaluated on MMEB-V1 and **30+ models** on MMEB-V2. The inclusion of these state-of-the-art models in **Table7** further validates that our benchmark has been broadly adopted for evaluating multimodal embedding models.
>
>
> >More detailed reporting of training resources, specifically the training time of VLM2Vec-V2 on 8 NVIDIA H100 GPUs and how its computational cost compares to baseline models.
>
> We have added specific training details to **Section 4.1.1**. Training VLM2Vec-V2 (2B) on **8 NVIDIA H100 GPUs** takes approximately **15 hours per 1k steps**. While many baselines do not report detailed training resource information—making a direct comparison difficult—we believe our focus on the 2B-parameter scale provides a highly efficient and reproducible training recipe for labs working under an academic budget.
>
>
> >Replace arXiv references with their corresponding peer-reviewed, published versions where available.
>
> We have thoroughly proofread the bibliography and replaced arXiv references with their corresponding peer-reviewed, published versions (e.g., COLM 2024, ICLR 2025, and ACL 2025).

---

> ### Author Response · Authors · 2025-12-24
> **Rebuttal by Authors (Part 2)**
>
> >Whether baseline approaches were trained on datasets included in the benchmark.
>
> For the baseline models reported in this paper, we have reviewed their public documentation and are confident they did not utilize the test splits included in our benchmark. For external models on the public leaderboard, we acknowledge that it is impossible to verify contamination for models that have not publicly disclosed their training recipes. However, we strictly ensure that all datasets in the MMEB-V2 benchmark consist exclusively of test splits from each task and are held out from our own training set.
>
>
> >The datasets included in the MMEB-v2 benchmark are completely distinct from the ones used to train the embedding model VLM2Vec-V2.
>
> Yes, all datasets included in the MMEB-V2 benchmark consist solely of test splits from each task and are strictly held out from training the VLM2Vec-V2 embedding model.
>
>
> >Criteria for deciding which datasets are included in the benchmark and the training set.
>
> When designing the benchmark, we follow two key criteria.
>
> (1) Diversity: the benchmark should be highly diverse, covering a wide range of tasks, domains, and modality combinations  (spanning 9 meta-tasks and 78 datasets).
>
> (2) Practicality: the benchmark requires a reasonable amount of computation and evaluation time, enabling users to efficiently iterate on and compare their models.
>
> For the training set, we consider public data with explicit training splits. The training set was optimized through iterative experimentation to find the most effective data mixture for broad cross-modal generalization.
>
>
> >Broader Impact
>
> We have added **Section7** to discuss the broader impacts and limitations of our work.

---

### Review · Reviewer_CKXA · 2025-12-04

**Summary Of Contributions:**

The paper proposes a new framework for learning embeddings for a variety of different visual forms and introduces an extended benchmark that adds five new task types to an existing benchmark. The paper reports the results of experiments on both the new tasks and the original benchmark tasks. The results demonstrate that the proposed approach outperforms (or performs competitively with) existing methods and performs well on the new task types.

**Audience:**

Yes

**Audience Explanation:**

The paper addresses an important area and makes an interesting contribution, contributing both an expanded benchmark with new tasks as well as the results of a new framework.

**Claims And Evidence:**

No

**Claims Explanation:**

It is claimed in the abstract and introduction that the paper:
(1)	proposes a framework (VLM2Vec-V2) for learning embeddings across diverse visual forms.
(2)	introduces an extended benchmark that adds five new task types to an existing benchmark.
(3)	demonstrates through extensive experiments that VLM2Vec-V2 achieves strong performance on the newly introduced video and document retrieval tasks and improves over prior baselines on the original image benchmarks.
(4)	offers insights into the generalizability of various multimodal embedding models.
(5)	highlights effective strategies for unified embedding learning.
(6)	lays the groundwork for more scalable and adaptable representation learning in both research and real-world settings.

Claim 1 is established through the introduction of a framework for fine-tuning a VLM. Claim 2 is established through the extension of the benchmark. Claim 3 is supported by extensive experiments, although the analysis and discussion of the experimental results could be much more thorough.

It is less clear that the paper satisfactorily achieves claims 4-6 and the claims should probably be moderated, phrased carefully, or made precise. The discussion of the main results is very brief and it is difficult to extract from the paper the “insights into the generalizability of … models”. Most of the commentary is dedicated to observing that the introduced model is competitive or outperforms other methods.
The paper provides an ablation analysis that studies performance on various modalities when the model is trained on different (combinations of) modalities, but there is no in-depth analysis of these results either. Claim 6 seems unsupported since there is no evidence that the results extend to “real-world” settings.

**Requested Changes:**

(1)	The title indicates that this is V2 of VLM2VEC. With this in mind, it is important to clarify what the main changes are compared to VLMVEC. The related work glosses over VLMVEC, mentioning it only in passing. While the paper doesn’t need to dwell on this too much, it is important for a reader to understand what have emerged as critical changes with the incorporation of videos and related tasks. The changes to the contrastive learning and data formatting appear minor, but more than half of the text in Section 3 is dedicated to them.

(2)	The discussion of the main results (Section 4.2) is very brief and does not amount to much beyond “our model performs well”. The paper would be strengthened by a considerably more detailed analysis. Insights into how some of the baseline models performed on particular tasks would be useful (when and why do they fail; when do they do well). Are there tasks where the introduced model performed slightly less well? If so, is there an understanding as to why this might be the case? Within the tasks themselves, do models tend to have the same failure cases (e.g., in MRET are >55 percent of the tasks failed by all models) or do the successes vary between models? There should be many ways to delve into the experimental results.

(3)	The ablation analysis results are interesting, but the experiment results are again collapsed to a single table without any nuanced or in-depth analysis. Counterintuitive results are not explained or discussed (why is performance on video the worst when the training source is video?). There are no confidence intervals provided, nor are is there any statistical significance testing, so it is not clear that the reported differences are meaningful. There are concerns that the results are not genuinely a reflection of the modality of the data but perhaps more the amount, quality, and variety of the data. It’s also unclear why Video training data would be so useful for VisDoc (considerably better than VisDoc itself).

---

> ### Author Response · Authors · 2025-12-28
> **Rebuttal by Authors (Part 1)**
>
> We sincerely appreciate the reviewer’s insightful feedback. In the revised manuscript, we have made the following improvements:
> * **Introduction and Abstract Revision:**
>  We refined the claims in the introduction and abstract to ensure they accurately reflect the contributions supported by our experiments.
> * **Clear Separation of Novel Contributions in Section 3:**
>  Section 3 has been reorganized for better clarity: Section 3.1 now provides a preliminary overview of the VLM2Vec framework. Section 3.2 discusses foundation model selection. Sections 3.3–3.4 introduce the key new technical contributions in VLM2Vec-V2.
> * **Expanded Analysis of Multimodal Training Configurations:**
>  Section 4.4.1 has been rewritten to provide deeper analysis of modality mixing, and we additionally include statistical significance testing in Section A.5.
> * **Qualitative Analysis:**
> We add a new Section 4.4 to provide qualitative analysis of the advantages of VLM2Vec-V2 over the baseline model. The corresponding examples are shown in Section A.6.
>
> **All updates have been clearly marked in blue for easy reference.**

---

> ### Author Response · Authors · 2025-12-28
> **Rebuttal by Authors (Part 2)**
>
> >Some claims in the introduction and abstract aren't precise.
>
> Thank you for the feedback. We have updated both the introduction and abstract to remove claims that were not sufficiently supported within the paper. Specifically, we removed the following points: (4) offers insights into the generalizability of various multimodal embedding models. (6) lays the groundwork for more scalable and adaptable representation learning in both research and real-world settings.
>
> Regarding (5) — highlights effective strategies for unified embedding learning — we have substantially revised Section 4.4.1 to provide a clearer and more thorough discussion of our key conclusions on multimodal training data mixing. The updated section now emphasizes three key findings:
>  * **Image supervision is a crucial foundation** for adapting MLLMs into unified multimodal embedding models.
>  * **Data source quality and modality compatibility critically influence generalization**: mixing image data with high-quality VisDoc supervision improves performance across modalities, whereas combining image with lower-quality or synthetic video data can degrade performance.
> * **Full Image + VisDoc + Video mixtures yield the best overall performance**, demonstrating that broader modality coverage is beneficial when additional modalities are carefully curated and image remains the anchor modality.
>
> To rigorously support these results, we additionally conducted a task-level paired t-test to analyze the statistical significance of different modality mixing strategies (Section A.5).
>
> >clearer articulation of the key differences between VLM2Vec-V2 and VLM2Vec
>
> We have reorganized Section 3 to clearly differentiate our new contributions from the original framework. Section 3.1 provides a preliminary overview of the VLM2Vec framework and Section 3.2 describes the foundation model selection, while Sections 3.3–3.4 detail the novel extensions in V2 . These include our unified multimodal data mixing strategy and the homogeneous sub-batching scheme designed to balance heterogeneous data sources and increase contrastive difficulty.
>
> >Provide a more detailed breakdown of ablation results.
>
> We thank the reviewer for this important insight. We agree that treating “Video” or “Document” as monolithic data sources is insufficient, and that data quality and diversity can be confounding factors. In response, we have substantially revised Section 4.4.1 (along with Tables 3 and 4) to provide more fine-grained comparisons and analyses in the updated manuscript. For all the experiments, we fix the training scale (100K samples).
>
> >Include statistical significance testing to determine whether observed performance differences are meaningful rather than noise.
>
> To rigorously assess the generalization impact of multimodal training configurations, we further conduct a task-level paired t-test to evaluate statistical significance (Section A.5). We conduct a task-level paired t-test to evaluate statistical significance.
>
> The below table is Task-level significance evaluation of different modality mixing strategies compared to the Image 200K baseline. Listed values indicate the mean performance change across 78 MMEB-V2 tasks, with positive values reflecting consistent gains. Statistical reliability is assessed using paired *t*-tests: * p < 0.05, ** p < 0.01, *** p < 0.001.
>
> | Training Data Configuration                         | Image Δ    | Video Δ   | VisDoc Δ     | Global Δ  |
> |-----------------------------------------------------|:----------:|:---------:|:------------:|:---------:|
> | Image 200K (Baseline)                                | –          | –         | –            | –         |
> | Image 100K + ColPali 100K                            | -1.74      | -0.02     | +7.76 ***    | +1.58 *   |
> | Image 100K + VideoCap 100K                           | -2.52 **   | -0.89     | -0.36        | -1.48 *** |
> | Image 100K + ColPali 50K + VideoCap 50K              | -1.55 **   | +1.09 *   | +6.69 ***    | +1.59 **  |
> | All Sources 200K                                     | -2.13 **   | -0.84     | +10.19 ***   | +1.96 *   |
>
> The significance results in the table confirm the key observations from Section 4.4.1. In particular, effective modality mixing requires both high-quality data and modality compatibility: adding curated VisDoc supervision (e.g., ColPali) leads to statistically significant gains, whereas incorporating noisier video supervision (VideoCap) can negatively affect performance. Furthermore, the two full Image + VisDoc + Video mixtures deliver the strongest global improvements, highlighting that broader modality coverage is most beneficial when additional modalities are carefully selected and curated.

---

> ### Author Response · Authors · 2025-12-28
> **Rebuttal by Authors (Part 3)**
>
> >Why Video training data would be so useful for VisDoc (considerably better than VisDoc itself)
>
> In the original submission, the comparison was affected by an imbalance in the amount of training data provided for each modality, which made the results difficult to interpret fairly. In the revised version, we conduct an ablation under a strictly controlled setting where each model is trained with the same amount of data. As shown in Table 3, under this matched-data setup, the counterintuitive trend no longer appears: the model trained on VisDoc data now clearly outperforms the model trained on Video data when evaluated on VisDoc tasks (61.72 vs. 41.55 on the VisDoc modality). This suggests that the earlier observation was primarily an artifact of imbalanced training data rather than an inherent advantage of Video supervision for VisDoc evaluation.
>
> >Why is performance on video the worst when the training source is video
>
> This counterintuitive observation partially persists even when we control the total amount of training data. As shown in Table 3, the effectiveness of Video-only training varies substantially by data source: compared to curated VisDoc data, current Video datasets are relatively noisy and offer weaker supervision, which can lead to degraded performance when used in isolation.
>
> However, Table 4 further shows that incorporating Video data **together** with Image and VisDoc data still contributes positively to the overall embedding quality. In other words, while Video data alone may harm performance, **its complementary information becomes beneficial when paired with strong image and document supervision**. This motivates our decision to retain Video signals in the final training configuration.
>
> We acknowledge the limitations of current Video training data and plan to explore better-curated video resources in future work to more effectively support multimodal embedding learning.
>
> >Qualitative analysis of when our model performs better or worse than baseline models
>
> We add a new Section 4.4 to provide qualitative analysis of the advantages of VLM2Vec-V2 over the baseline model. The corresponding examples are shown in Section A.6.
>
> We perform qualitative comparisons between VLM2Vec-V2 and VLM2Vec on representative examples spanning both video and document tasks. On video tasks, VLM2Vec-V1 mainly relies on surface-level visual or keyword similarity rather than understanding the details of the entire video and the given text. In contrast, VLM2Vec-V2 leverages visual context across all frames, allowing it to locate the correct moment that aligns with the intended meaning of the query, demonstrating a more comprehensive video representation. For document understanding, VLM2Vec-V1 primarily depends on keyword matching rather than reading and understanding the full content of the document. In contrast, VLM2Vec-V2 considers the broader semantic information and identifies the correct article that contains the detailed content required by the query. This demonstrates that VLM2Vec-V2 offers a stronger capability in document comprehension beyond superficial keyword overlap.

---

### Author Response · Authors · 2025-12-28
**​​Summary of Revision**

We thank the reviewers for their time and expertise in evaluating our work, and we are grateful for the constructive feedback. To better address the questions and concerns raised and to further improve the paper’s quality, we have made the following updates to the revised PDF. These updates are also reflected in our responses to the reviewers. **All updates have been clearly marked in blue for easy reference.**

* **Introduction and Abstract Revision:**
 We refined the claims in the introduction and abstract to ensure they accurately reflect the contributions supported by our experiments.
* **Clear Separation of Novel Contributions in Section 3:**
 Section 3 has been reorganized for better clarity: Section 3.1 now provides a preliminary overview of the VLM2Vec framework. Section 3.2 discusses foundation model selection. Sections 3.3–3.4 introduce the key new technical contributions in VLM2Vec-V2.
* **Expanded Analysis of Multimodal Training Configurations:**
 Section 4.4.1 has been rewritten to provide deeper analysis of modality mixing, and we additionally include statistical significance testing in Section A.5.
* **Qualitative Analysis:**
We add a new Section 4.4 to provide qualitative analysis of the advantages of VLM2Vec-V2 over the baseline model. The corresponding examples are shown in Section A.6.
* **Clearly specifying the exact evaluation metrics used for each task:**
We have added Section 4.2 to explicitly define the metrics for each modality .
* **Include SOTA models evaluated on our benchmarks:**
Appendix A.4 has been updated to incorporate the most recent leaderboard results as of December 2025. Since the release of the MMEB series, it has become a standard evaluation in the domain; as of late 2025, over 50 models have been evaluated on MMEB-V1 and 30+ models on MMEB-V2.
* **Broader Impact:**
We have added Section7 to discuss the broader impacts and limitations of our work.

---

### Decision · Action_Editor_i8Rc · 2026-01-15

**Recommendation:** Accept as is

**Audience:**

Yes

**Audience Explanation:**

This will be of interest to a reasonably broad community in the TMLR audience. The paper provides a valuable benchmark and a thorough empirical study of multimodal embedding models across a wide range of tasks and modalities. These contributions will be useful to researchers working on multimodal and representation learning, and the paper is likely to serve as a useful reference and resource for the community.

**Claims And Evidence:**

Yes

**Claims Explanation:**

This paper presents a comprehensive benchmark together with a unified multimodal embedding framework spanning images, videos, and visual documents. Across all three reviews, there is clear agreement that the experimental evaluation is extensive and convincing, and that the revised manuscript has appropriately moderated its claims while strengthening the analysis in response to reviewer feedback. The authors have improved methodological clarity, added targeted analyses, and refined the presentation, resulting in a technically sound and well-supported contribution. Overall, the evidence presented supports the claims made in the paper, and the revisions have substantially strengthened the work. I therefore recommend acceptance.